# Extreme melting at Greenland's largest floating ice tongue

**Ole Zeising**[1,a], **Niklas Neckel**[1], **Nils Dörr**[3,b], **Veit Helm**[1], **Daniel Steinhage**[1], **Ralph Timmermann**[1], **and Angelika Humbert**[1,2]

[1]Alfred-Wegener-Institut Helmholtz-Zentrum für Polar- und Meeresforschung, Bremerhaven, Germany
[2]Department of Geosciences, University of Bremen, Bremen, Germany
[3]Institute of Photogrammetry and Remote Sensing, Karlsruhe Institute of Technology (KIT), Karlsruhe, Germany
[a]formerly at: Department of Geosciences, University of Bremen, Bremen, Germany
[b]formerly at: Institute of Geosciences, Kiel University, Kiel, Germany TS1

**Correspondence:** Ole Zeising (ole.zeising@awi.de)

**Abstract.** The 79° North Glacier (Nioghalvfjerdsbrae, 79NG) is one of three remaining glaciers with a floating tongue in Greenland. Although the glacier has been considered exceptionally stable in the past, earlier studies have shown CE1 that the ice tongue has thinned in recent decades. By conducting high-resolution ground-based and airborne radar measurements in conjunction with satellite remote-sensing observations, we find significant changes in the geometry of 79NG. In the vicinity of the grounding line, a 500 m high subglacial channel has grown since ∼ 2010 and has caused surface lowering of up to $7.6\,\mathrm{m\,a^{-1}}$. Our results show extreme basal melt rates exceeding $150\,\mathrm{m\,a^{-1}}$ over a period of 17 d within a distance of 5 km from the grounding line, where the ice has thinned by 32 % since 1998. We find a heterogeneous distribution of melt rates, likely due to variability in water column thickness and channelization of the ice base. Time series of melt rates show a decrease in basal melting since 2018, indicating an inflow of colder water into the cavity below 79NG. We discuss the processes that have led to the changes in geometry and conclude that the inflow of warm ocean currents has led to the extensive thinning of 79NG's floating ice tongue near the grounding line over the last 2 decades. In contrast, we hypothesize that the growth of the channel results from increased subglacial discharge due to a considerably enlarged area of summer surface melt due to the warming of the atmosphere.

## 1 Introduction

The mass loss of the Greenland Ice Sheet over the last decades as a result of a warming atmosphere and ocean has accelerated (Shepherd et al., 2020) and has contributed to recent sea-level rise by $1.4\,\mathrm{mm\,a^{-1}}$ (Khan et al., 2022a). Half of the mass loss is caused by ice-sheet discharge through marine-terminating glaciers (Shepherd et al., 2020), mainly due to the retreat of glacier fronts (King et al., 2020) because the floating ice tongues restrain the outflow of the grounded ice (Fürst et al., 2016). The largest of the three remaining floating tongues in Greenland is that of the 79° North Glacier (Nioghalvfjerdsbrae, 79NG). Together with its neighboring Zachariae Isstrom (ZI), it is the main outlet glacier of the Northeast Greenland Ice Stream (NEGIS; Fig. 1a), the largest ice stream of the Greenland Ice Sheet (Fahnestock et al., 2001). After the collapse of ZI's floating tongue in 2002, the glacier itself (Khan et al., 2014; Mouginot et al., 2015), as well as the NEGIS, has shown an extensive speedup with respect to ice flow velocity (Khan et al., 2022b). In contrast, only minor ice flow velocity acceleration has been observed at 79NG (Mouginot et al., 2015; Vijay et al., 2019). CE2

Ice-sheet simulations indicate that 79NG remains stable within this century and will experience only a minor grounding line retreat as bedrock rises inland (Choi et al., 2017). Its stability is attributed to pinning points at the calving front (Thomsen et al., 1997), lateral resistance from shear margins (Mayer et al., 2000; Rathmann et al., 2017; Mayer et al., 2018), and confinement of the glacier that leads to lateral compression. However, thinning has occurred over the last

2 decades (Helm et al., 2014; Kjeldsen et al., 2015; Mouginot et al., 2015; Mayer et al., 2018), and cracks have formed at the calving front that might be a precursor to disintegration (Humbert et al., 2023).

Observations and modeling show that the inflow of warm Atlantic Intermediate Water (AIW, with temperatures exceeding 1 °C) into the cavity below 79NG (Straneo et al., 2012; Wilson and Straneo, 2015; Lindeman et al., 2020; Schaffer et al., 2020) and its variability are connected to the ocean currents in Fram Strait (Münchow et al., 2020; von Albedyll et al., 2021). Based on temperature and salinity measurements in an epishelf lake, Bentley et al. (2023) showed that AIW reaches the grounding line area of 79NG a few months after entering the cavity. The observed oceanic heat transport into the sub-ice cavity (Schaffer et al., 2020) has been suggested to maintain intense basal melting (Mayer et al., 2018; Lindeman et al., 2020; Schaffer et al., 2020). Similar processes might have led to the disintegration and retreat of the floating ice tongues of Jakobshavn Glacier (Motyka et al., 2011). In the future, basal melt rates are expected to show the most pronounced increase in the northeastern part of Greenland towards the end of the 21st century (Slater et al., 2020). However, the supply of fresh water from glacial surface melting has been found to alter circulation in fjords as well as the basal melting of glaciers by increasing buoyancy-driven circulation and decreasing shelf-forced circulation (Straneo et al., 2016). Subglacial water discharge from beneath the grounded ice is often linked to the location of basal channels in the floating ice shelves caused by locally enhanced melting (Le Brocq et al., 2013). Such channels can be up to a few kilometers in width and up to a few hundred meters in height (Rignot and Steffen, 2008).

The spatial distribution of basal melt rates can be investigated using repeated measurements with a phase-sensitive radio echo sounder (pRES). The same device can be operated in an autonomous mode (henceforth ApRES) to perform measurements over a longer period of time with a defined interval. Previous studies used pRES and ApRES measurements to investigate the spatial distribution and temporal variability in basal melting inside basal channels. For example, at the Ross Ice Shelf, Antarctica, Marsh et al. (2016) found enhanced melting inside a channel near the grounding line which decreased in the downstream direction. For a channel at Filchner Ice Shelf, Antarctica, Humbert et al. (2022) revealed that melt rates inside the channel decreased in the direction of ice flow and fell below those outside the channel, causing the channel height to decrease. While they found no pronounced seasonality in melting inside the channel, Washam et al. (2019) detected a significant increase in melting inside a channel at Petermann Glacier, Greenland, during the surface melt period in summer. They linked the seasonality to the increased subglacial discharge that enhanced the inflow of warmer ocean currents into the cavity (Shroyer et al., 2017; Washam et al., 2019). Whether basal channels stabilize or weaken shelf ice is not yet fully understood (Alley et al., 2016). Numerical models indicate that the existence of channels can decrease the mean basal melt rate (Millgate et al., 2013); at the same time, the channels can also structurally weaken the ice shelf (Vaughan et al., 2012).

Observations of basal melt rates and their influence on ice thickness are considered key to understanding the dynamics of the system. Basal melt rates for 79NG have been estimated based on indirect satellite remote-sensing retrievals (Wilson et al., 2017), which are accompanied by considerable uncertainties. Particularly within a few kilometers of the grounding line and above basal channels, the ice is in hydrostatic imbalance, limiting the analysis of melt rates based on changes in surface elevation (Chartrand and Howat, 2023). Thus, other methods must be used to monitor changes in ice thickness and to understand the underlying processes, especially in the area of the grounding line of 79NG where higher basal melt rates are expected due to thick (reduced melting temperature) ice coming into contact with warm ocean waters.

In this study, we investigate the recent changes in the ice thickness of 79NG using in situ observations, airborne observations, and satellite remote-sensing observations. We analyze the spatial distribution of thinning and basal melt rates focusing on the vicinity of the grounding line of 79NG. Finally, we discuss the processes that explain the observations and how these have changed in the past decades.

## 2   Data

In order to obtain a time series of surface elevations of the 79NG grounding line area, we generated 96 digital elevation models (DEMs) from bistatic TanDEM-X SAR (synthetic aperture radar) interferometry that span the period from December 2010 to April 2021. Additionally, we acquired airborne and ground-based radar measurements at 79NG within the framework of the "Greenland Ice Sheet–Ocean Interaction" (GROCE, https://www.groce.de, last access: 1 March 2024) project. The airborne radar measurements were performed in April 2018 and July 2021 with the AWI (Alfred Wegener Institute, Helmholtz Centre for Polar and Marine Research) ultra-wideband (UWB) radar in order to determine the basal geometry of 79NG. We obtained the spatial distribution of Lagrangian thinning rates from a repeat survey of pRES measurements in July 2017 and 2018. In July 2017, we marked the measurement location on the surface with 4 m long bamboo stakes drilled into the ice in order to be able to repeat the measurement in 2018 at exactly the same location in the Lagrangian frame. The majority of the measurement locations were distributed within an 8 km distance of the grounding line (Fig. 1b). Here, we operated ApRES stations at three locations (ApRES1–ApRES3) until September 2023; these ApRES moved with the ice to derive year-round time series of basal melt rates in a Lagrangian reference frame. In summer 2018, we relocated ApRES2 to its starting position from 2016 in order to repeat the mea-

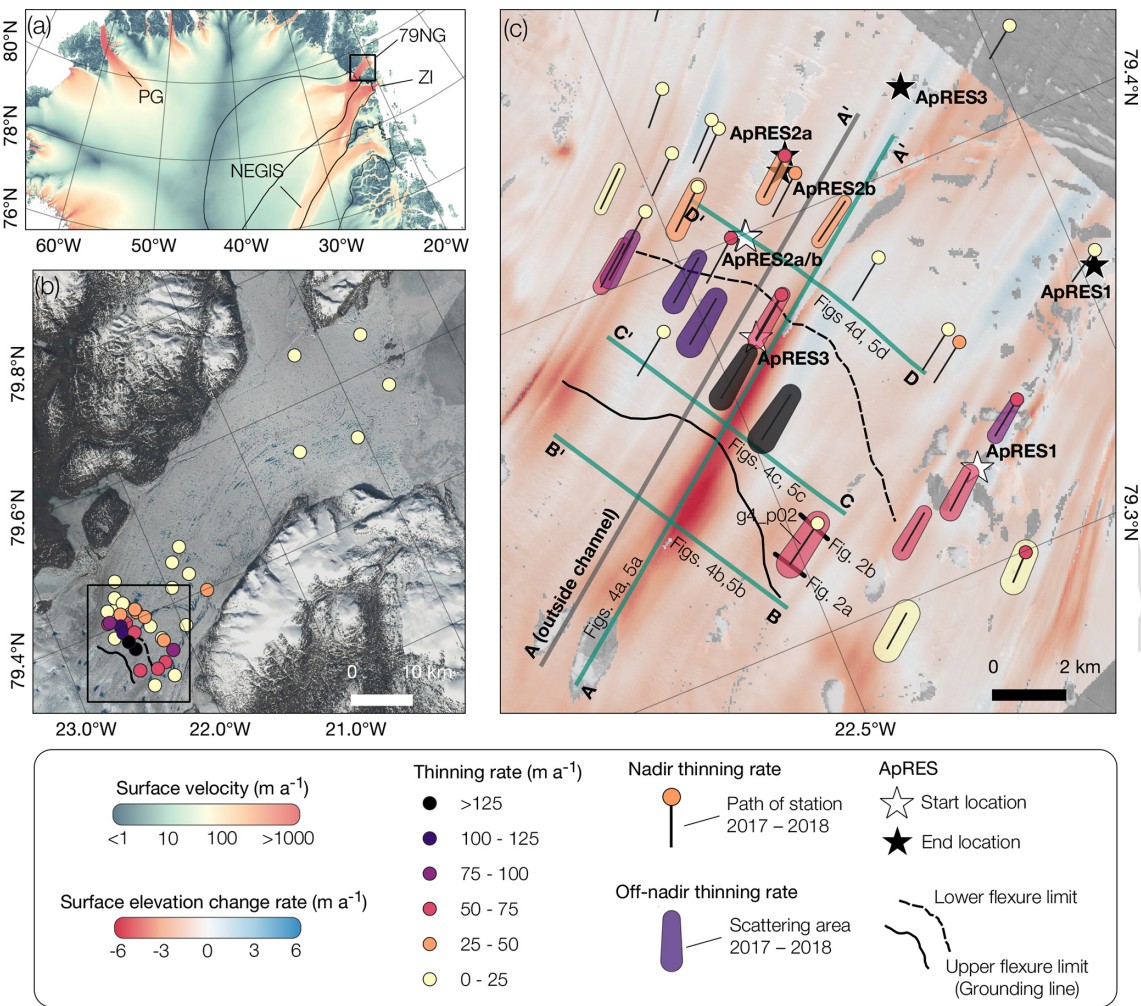

**Figure 1. (a)** Map of northern Greenland with the drainage basins (black lines) of 79NG and Zachariae Isstrom (ZI) (Krieger et al., 2020) and surface velocities (Joughin et al., 2018) showing the Northeast Greenland Ice Stream (NEGIS) and Petermann Glacier (PG). **(b)** A Sentinel-2 mosaic of 79NG with thinning rates derived from pRES measurements in 2017 and 2018 (box in panel **a**). **(c)** Enlargement of the 79NG hinge zone (box in panel **b**) with surface elevation change rates (d$h$/d$t$) derived from TanDEM-X satellite data between 2010 and 2021. Dots and scattering areas show nadir and off-nadir thinning rates with paths of the Lagrangian pRES measurement location between July 2017 and July 2018. White stars mark the starting position and black stars mark the ending position of ApRES stations. The upper flexure limit (grounding line) and lower flexure limit are based on interferometry and mark the area where the ice is being bent by the tides. Copernicus Sentinel data from 2018 were retrieved from the Copernicus SciHub on 16 August 2021.

surements on the same flowline. These stations are labeled as ApRES2a (2016–2018) and ApRES2b (2018–2019).

## 3 Methods

### 3.1 Time series of surface elevations from TanDEM-X SAR interferometry

DEMs were generated from bistatic TanDEM-X SAR interferometry closely following the methods described by Neckel et al. (2013). Interferograms were formed from co-registered single-look slant range complex (CoSSC) data employing a multi-looking step with four looks in both the range and azimuth directions. Prior to phase unwrapping, we subtracted a simulated phase from the global TanDEM-X DEM at a 30 m resolution (Wessel et al., 2016). The latter was done to reduce unwrapping errors, and the simulated phase was added back afterwards. The final DEMs were geocoded and spatially adjusted to the global TanDEM-X DEM by calculating the standard deviation and the normalized median absolute deviation (NMAD) for all DEMs over stable ground (Nuth and Kääb, 2011; Wessel et al., 2016). The NMAD is considered to be more robust to outliers than the standard deviation (Höhle and Höhle, 2009). For the entire stack of DEMs, we obtain a standard deviation of 0.96 m and an NMAD of 0.55 m, which is in agreement with the

TanDEM-X mission requirements (Wessel et al., 2018). Surface elevation changes between 2010 and 2021 were estimated by fitting a linear trend to every pixel of the co-registered stack of 96 DEMs (e.g., Berthier et al., 2016).

## 3.2 Ultra-wideband (UWB) airborne radar

The UWB is a multichannel coherent airborne radar that consists of an eight-element antenna array with a total transmission power of 6 kW (Hale et al., 2016). The antennas operate in the frequency band of 150–520 MHz, with a pulse repetition frequency of 10 kHz and a sampling frequency of 1.6 GHz. The characteristics of the transmitted waveform and the recording settings can be manually adjusted. We used alternating sequences of different transmission–recording settings (waveforms) to increase the dynamic range: short pulses (1 μs) and low receiver gain (11–13 dB) to image the glacier surface; longer pulses (3–10 μs) with higher receiver gain (48 dB) to image internal features and the ice base. The waveforms were defined with regard to the glacier thickness. Additionally, we used two different frequency bands in the survey: 180–210 and 150–520 MHz. The theoretical range resolution in ice after pulse compression for the two bandwidths is about 2.8 and 0.23 m, respectively. Recorded traces were pre-stacked in the hardware by a factor of between 2 and 16, depending on the pulse length. In order to reduce range side lobes, the transmitted and the received signals were tapered using a Tukey window, while the received signal spectrum was filtered with a Hanning window (The MathWorks Inc., 2022). We recorded the position of the aircraft with four NovAtel GPS receivers, which were mounted on the wings and the fuselage.

Post-flight processing included pulse compression in the range direction, SAR focused in the along-track direction, and array processing to increase the signal-to-noise ratio and to suppress off-nadir echoes. The SAR focus is set to achieve a ground resolution of 10 m in the along-track direction. To transform two-way travel time to depth, we used a propagation velocity for the electromagnetic wave of 168.914 m μs$^{-1}$, which refers to a relative permittivity of $\varepsilon_r = 3.15$ for pure ice. No firn correction was applied, as the predominant part of the glacier is located in the ablation zone. We concatenated the echograms of the alternating waveforms to obtain the final echograms covering the glacier from the surface to the base with a high dynamic range. Finally, the surface return of the radar echo was aligned with a high-resolution elevation model with a vertical accuracy of 0.1 m, which was determined from simultaneously acquired laser scanner data.

## 3.3 Phase-sensitive radio echo sounder (pRES)

### 3.3.1 Technical background

The pRES is a ground-penetrating, frequency-modulated continuous-wave (FMCW) radar that transmits chirps with a frequency bandwidth of 200 MHz and a center frequency of 300 MHz (Brennan et al., 2014; Nicholls et al., 2015). While the repeat pRES survey was performed with 100 chirps and 2 skeleton slot antennas, the ApRES stations consisted of 2 bow-tie antennas and recorded 20 chirps with a measuring interval of between 1 and 6 h. For processing the raw data, we calculated pairwise correlation coefficients of all chirps, rejected chirps with low correlation coefficients, and stacked the remaining ones. We followed the processing methods outlined in Brennan et al. (2014) to get amplitude- and phase-depth profiles. We assumed a relative permittivity of $\varepsilon_r = 3.15$ in ice for the time-to-depth conversion.

### 3.3.2 Thinning rates from single repeated pRES measurements

The estimation of the Lagrangian thinning rate is based on the change in ice thickness along the flow of the same ice particles. The ice base is assumed to be responsible for strong peaks in the radar signal due to the high contrast in relative permittivity between ice and seawater. In the case of a flat ice base, the nadir reflection has the shortest two-way travel time of all basal reflections in a radius defined by the antenna beamwidth. However, steep basal gradients, such as those of basal channels, can cause off-nadir reflections that might appear before the nadir basal return. On average, the steeper the basal gradient between the nadir ice base and the location of the off-nadir reflection increases, the earlier the off-nadir reflection occurs. Thus, whether or not an off-nadir reflection appears before the nadir basal return depends on the location of the measurement relative to the surrounding basal slopes and their gradients.

In order to identify nadir and off-nadir returns, we used the first multiple reflections from the ice base, which were characterized by twice the two-way travel time because they originated from the reflections at the ice base, the ice surface, and again at the ice base. Here, we assume that the multiple is strongest for the nadir reflection, as most of the reflected energy from a far-off-nadir reflection is reflected in the opposite direction in the case of a flat ice surface. Therefore, multiples from off-nadir reflections will be weaker compared with nadir reflections. Additionally, we used the ice thickness distribution derived from UWB echograms near the location of the pRES observations, as they can reveal the ice thickness and provide information on the origin of the recorded off-nadir reflection. At pRES location g4_p02, UWB echograms from 2018 show a growing subglacial channel in the immediate vicinity of both the first and repeated pRES measurement (Figs. 1c, 2). Based on these UWB echograms, we link

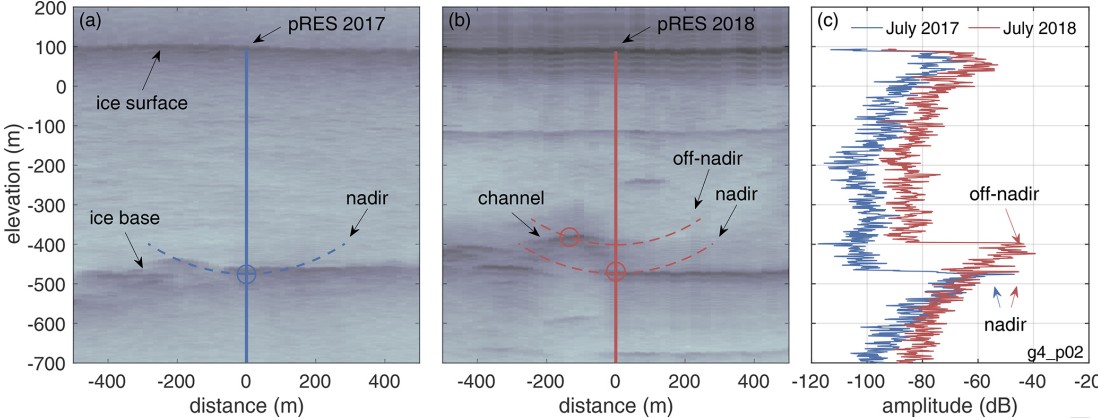

**Figure 2.** Growing basal channel from pRES and UWB echograms. **(a, b)** UWB echograms from the across-flow profiles from 2018; the center of both is the location of a Lagrangian pRES measurement in 2017 (vertical blue line in **a**) and 2018 (vertical red line in **b**). Possible origins of nadir and off-nadir reflections, discovered in the pRES echograms **(c)**, are represented by dashed lines. The suggested locations at which the reflections occurred are marked by circles. **(c)** pRES echograms from 2017 and 2018 with the identified nadir and off-nadir reflections. The location of the profiles shown in panels **(a)** and **(b)** is shown in Fig. 1c. CE4

the origin of the off-nadir reflection to the basal channel. In Appendix A, we give further examples of amplitude profiles from repeated pRES measurements where the first basal return was identified to be an off-nadir return. The distinction between nadir and off-nadir returns is important, as the precise local change in ice thickness can only be revealed from nadir returns. Following Stewart et al. (2019), we applied a cross-correlation of the amplitude and the phase of their basal segments ranging from $-9$ to $+1$ m around the identified return. The uncertainty in this case is below 0.01 m.

Even if no distinction between nadir and off-nadir reflection can be made, the ice thickness change can be estimated with the approach outlined in the following. Using the first basal return in both measurements would always result in an underestimation of the change in ice thickness, at least at one of both locations where the (off-nadir) first basal reflections occurred (Appendix A). This means that a thinning rate exists somewhere in the scattering area that is equal to or even larger than the aforementioned estimated change in ice thickness CE5. Note that this should not be understood as the minimum rate in the scattered area (marked in Fig. 1 for off-nadir thinning rates), as there can also be lower thinning rates at the same time. Thus, at stations where we could not distinguish reliably between a nadir and an off-nadir reflection, we used the first strong increase in amplitude for the ice thickness calculation and interpreted this as an off-nadir return. The distance to CE6 the off-nadir basal reflector differs from the ice thickness above the reflector, which can be derived by $H^{\alpha} = R \cos \alpha$, where $R$ is the range of the basal reflector and $H^{\alpha}$ is the off-nadir ice thickness viewed at an angle $\alpha$. The resulting thinning rate $\Delta \dot{H}^{\alpha}$ (with positive values corresponding to thinning) is as follows:

$$\Delta \dot{H}^{\alpha} = -\frac{(R_2 - R_1) \cos \alpha}{\Delta t}, \tag{1}$$

where $R_1$ is the distance to the off-nadir basal reflector of the first measurement and $R_2$ is the distance to the off-nadir basal reflector of the second measurement after the time period $\Delta t$. As the off-nadir angle $\alpha$ is often unknown, we assumed that it ranged from 0° to a maximum of 30° (Brennan et al., 2014) and calculated the average of both angles. The spread in the ice thickness difference from both $\alpha$ and the inaccuracy in the signal propagation speed in the ice of $\sim 1\%$ (Fujita et al., 2000) is represented in the uncertainty of $\Delta \dot{H}^{\alpha}$. At those stations at which we identified nadir and off-nadir reflections, we determined both thinning rates. We rejected those measurements where the depth of the first basal return was unclear. The estimation of vertical strain was not possible with single repeated pRES measurements CE7 due to the low correlation of the amplitude profiles. We attributed this to water-saturated layers on the surface whose reflections also overlap with those of deeper layers. Thus, we could not calculate the basal melt rate.

### 3.3.3 Basal melt rates from ApRES time series

The calculation of basal melt rates follows previously described methods (Corr et al., 2002; Jenkins et al., 2006; Stewart et al., 2019). Several quantities cause changes in the range $R$ to the basal reflector within the time period $\Delta t$, including ablation $\Delta R_s$, strain $\Delta R_{\varepsilon}$, and basal melting $\Delta R_b$:

$$\frac{\Delta R}{\Delta t} = \frac{\Delta R_s}{\Delta t} + \frac{\Delta R_{\varepsilon}}{\Delta t} + \frac{\Delta R_b}{\Delta t}. \tag{2}$$

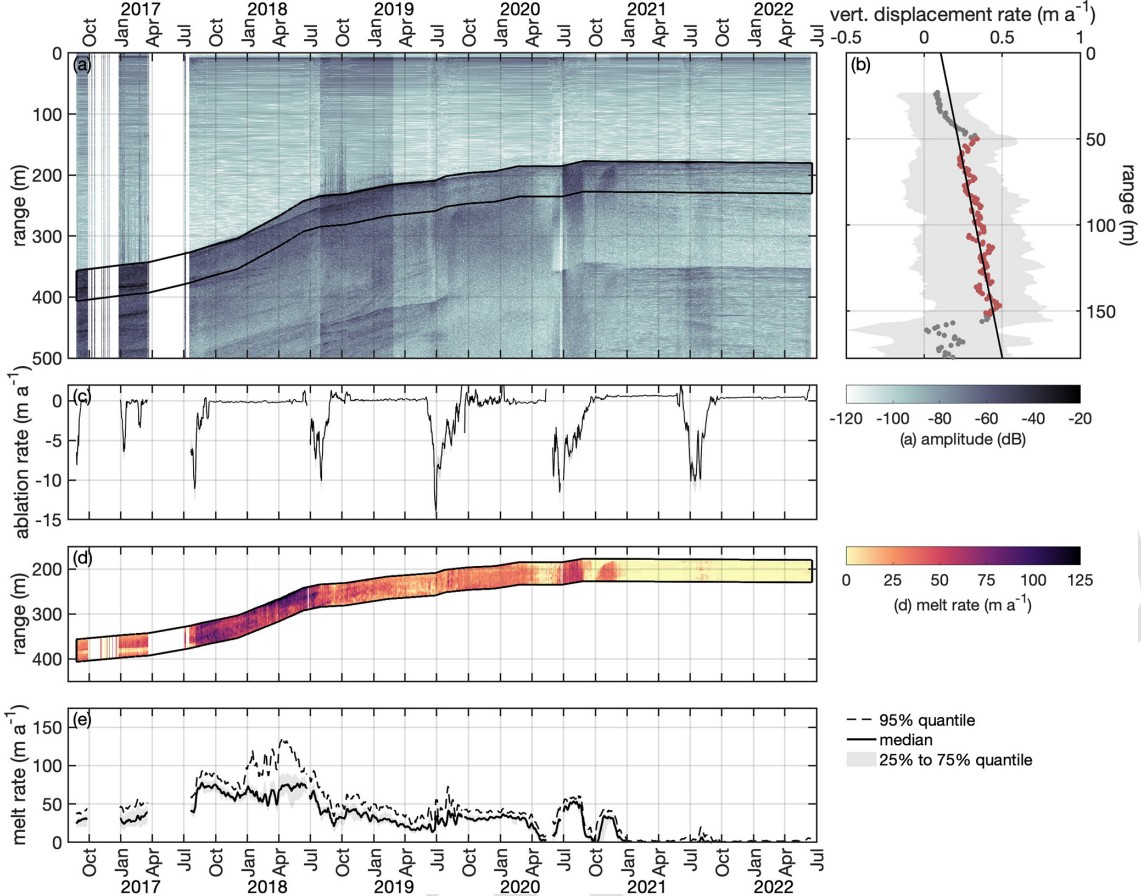

**Figure 3.** Analysis of ApRES1 time series. **(a)** Time echogram CE8 of a Lagrangian measurement at ApRES1 recorded between August 2016 and June 2022. In 2016 and 2017, several ApRES malfunctions caused data gaps. The black outline marks the first 50 m below the basal return. **(b)** Mean vertical displacement of englacial segments (dots). The gray shaded area marks the range between the 25 % and 75 % quantiles. Segments between 20 m below the surface and 20 m above the first basal return at the end of the measurement period (red dots) were used to calculate the change in ice thickness due to vertical strain by fitting a linear function (black line). **(c)** Time series of ablation rate (negative for ablation). The gray shaded area marks the uncertainty due to the off-nadir correction. **(d)** Time series of the determined melt rate (color) within the first 50 m below the basal return, corresponding to the area marked by black lines in panel **(a)**. **(e)** Time series of basal melt rate. The dashed line shows the 95 % quantile, the solid line shows the median, and the shaded area marks the range between the 25 % and 75 % quantiles.

With the ApRES time series (Fig. 3a, Appendix B), all of these quantities can be estimated in order to obtain the basal melt rate. As the estimation is based on the detection of the vertical displacement of layers, we divided the first echogram into 6 m long segments with 5 m overlap starting at a depth of 20 m. For each segment, we derived displacements from the complex cross-correlation of the phase of all pairwise time-consecutive measurements (Stewart, 2018). Afterwards, we calculated the daily mean values of the displacements.

In the first step, we used the temporal-mean vertical displacement of internal reflectors to calculate the vertical strain profile (Fig. 3b). Here, only those segments between 20 m below the surface and 20 m above the basal return at the last measurement were considered. In addition, we only considered measurements between October and May to avoid the

influence of ablation on the calculation of the strain. The vertical strain is the depth derivative of the vertical displacement $u_z$,

$$\varepsilon_{zz} = \frac{\partial u_z}{\partial z}, \tag{3}$$

which we derived from a linear fit that best matches the vertical displacements. Although one of the ApRES stations was located within the hinge zone in which bending might affect the strain distribution (Jenkins et al., 2006), none of the displacement distributions indicated a deviation from a linear function over depth.

For a nadir basal reflection, the estimation of the range shift due to ice deformation $\Delta R_\varepsilon^n$ is only affected by the vertical strain,

$$\Delta R_\varepsilon^n = \int_{0\,\mathrm{m}}^{R} \varepsilon_{zz}\,\mathrm{d}z, \tag{4}$$

from the surface at $z = 0\,\mathrm{m}$ to the ice base at $z = R$. Calculating the displacement of off-nadir reflectors due to deformation is more complex because the two horizontal strain components $\varepsilon_{xx}$ and $\varepsilon_{yy}$ must also be considered (see Appendix B1). As we can only determine the vertical strain component $\varepsilon_{zz}$ with ApRES measurements, we have to make assumptions to estimate $\Delta R_\varepsilon$ viewed at an angle $\alpha$. In Appendix B1, we show that for small off-nadir angles of $\alpha \leq 30°$, the absolute value of $\Delta R_\varepsilon$ is always less than or equal to the absolute value of the nadir displacement $\Delta R_\varepsilon^n$:

$$0 \leq |\Delta R_\varepsilon| \leq |\Delta R_\varepsilon^n|. \tag{5}$$

At all ApRES sites, we found $\varepsilon_{zz} > 0$ (see Fig. 3b and Appendix B2), so that Eq. (5) can be simplified further to

$$0 < \Delta R_\varepsilon \leq \Delta R_\varepsilon^n. \tag{6}$$

Thus, assuming that the reflection occurred from a nadir reflector, we cannot underestimate the deformation. The largest $\Delta R_\varepsilon^n$ was found to be 2.7 m for $\Delta t = 1\,\mathrm{year}$ CE9 at ApRES2b. If the change in ice thickness is based on an off-nadir basal reflection, the correction with the nadir range shift due to ice deformation underestimates the melt rate by $\leq 2.7\,\mathrm{m\,a^{-1}}$.

Next, we use the displacement time series of the segment centered at a range of $50\,\mathrm{m}$ ($u_z^{50}$) to correct for ablation (Fig. 3c). As the ice above is affected by ice deformation, we subtract this contribution from the displacement:

$$\Delta R_s^n = u_z^{50} - \int_{0\,\mathrm{m}}^{50\,\mathrm{m}} \varepsilon_{zz}\,\mathrm{d}z. \tag{7}$$

Here, $\Delta R_s^n$ is the vertical (nadir) displacement caused by ablation (negative for ablation). In order to determine the contribution of the ablation to the range difference for an off-nadir reflector, $\Delta R_s^n$, we need to correct for the angle $\alpha$:

$$\Delta R_s = \frac{\Delta R_s^n}{\cos\alpha}. \tag{8}$$

As $\alpha$ is still unknown, we use the extremes 0 TS2 and 30° and average both values. The difference from the mean is used as the uncertainty, which corresponds to up to $2\,\mathrm{m\,a^{-1}}$ in summer and near zero in winter. We removed outliers defined by ablation rates $> 0.1\,\mathrm{m\,d^{-1}}$.

To finally derive the basal melt rate $a_b$ in the vertical direction, we subtract $\Delta R_\varepsilon$ and $\Delta R_s$ from the displacement of a basal reflector:

$$a_b = -\frac{\Delta R_b}{\Delta t} = -\frac{\Delta R - \Delta R_s - \Delta R_\varepsilon}{\Delta t}. \tag{9}$$

Similar to Vaňková et al. (2021), we analyze $\Delta R$ for all segments within a range of $50\,\mathrm{m}$ below the first basal return to obtain the nadir and off-nadir basal melt rates (Fig. 3d). This range was chosen because all strong basal reflections occurred within the top $50\,\mathrm{m}$ at the ApRES sites. To represent the variability within a time series, we calculated the median melt rate as well as the 25 %, 75 %, and 95 % quantiles for each time step. Afterwards, a 7 d moving average filter was used to smooth the time series (Fig. 3e).

The largest uncertainty in the melt rate estimate arises from the unknown off-nadir angle $\alpha$, which affects the ablation and strain correction. The sum of both uncertainties is up to $8\,\mathrm{m\,a^{-1}}$ in summer, whereas it is significantly less in winter. In addition, the estimate of the melt rate quantifies the rate at which the ice base has approached the ApRES through melting. This can differ from the melt rate in the normal or vertical direction at the basal reflector. In the case of a flat ice base, the measured nadir melt rate is equal to the melt rate in the normal and vertical direction. For an inclined ice base, the measured nadir melt rate is equal to the melt rate in the vertical direction, which is different from that in the normal direction. A measured off-nadir melt rate can differ from the melt rate in both the normal and vertical directions. A further uncertainty arises from the inaccuracy in the signal propagation speed in the ice resulting in an inaccuracy in the melt rate of $\sim 1\,\%$ (Fujita et al., 2000).

## 4 Results

### 4.1 Growing subglacial channel causes local surface lowering

The DEM time series reveals that the surface elevation of 79NG has decreased along the grounding line at a rate of $-2.0 \pm 1.4\,\mathrm{m\,a^{-1}}$ (mean value $\pm$ standard deviation), corresponding to a decrease in the surface of $-20 \pm 14\,\mathrm{m}$ between December 2010 and April 2021 (Fig. 4a). The maximum surface-lowering value of $-56.9 \pm 0.1\,\mathrm{m}$ (or $-5.5 \pm 0.1\,\mathrm{m\,a^{-1}}$) for the same time period is evident in a graben-like structure in the center of the grounding line (Fig. 4a). This remarkable area of enhanced lowering is located downstream of a supraglacial lake and extends from $4\,\mathrm{km}$ upstream to $4\,\mathrm{km}$ downstream of the grounding line. Its width decreases in the flow direction from a maximum of $1\,\mathrm{km}$ roughly $2\,\mathrm{km}$ upstream of the grounding line to $500\,\mathrm{m}$ within $5\,\mathrm{km}$ of the grounding line in the ice flow direction (Fig. 1c). While a hill was present in the ice surface in the central area close to and upstream of the grounding line until 2015, this turned into a depression (Fig. 4b, c) due to enhanced surface-lowering rates compared with those rates outside the graben-like structure (Fig. 4e, f). The average elevation change upstream of the grounding line was $-2.1 \pm 0.1\,\mathrm{m\,a^{-1}}$ between December 2010 and April 2015; this value had increased to $-6.6 \pm 0.1\,\mathrm{m\,a^{-1}}$ by the end of the time series in April 2021

(Fig. 4b, e). Outside of this area, the surface subsided at a significantly slower rate of $-1.1 \pm 0.1\,\mathrm{m\,a^{-1}}$. A total of 5 km downstream of the grounding line behind the lower flexure limit where the ice floats freely (Fig. 1d), this sink already existed in 2010 and the surface-lowering rate was slower (Fig. 4d, g). At both locations downstream of the grounding line, the (Eulerian) surface elevation change rate suddenly changed in late 2019: it became less strong (Fig. 4f) and even turned into thickening (Fig. 4g), similar to what we observed until 2013.

In order to investigate what caused this drop in surface elevation, we recorded flight profiles with the UWB airborne radar. Near the grounding line, these airborne radargrams reveal the existence of several subglacial channels (Fig. 5 and Appendix Figs. C1 and C2). The largest channel by far, with a height of 500 m and a width of 1 km, is found in the central flow line near the grounding line and extends 5 km upstream from the grounding line. Above this channel, only 190 m of ice is left, which is 30 % of the surrounding ice thickness. The location of this channel is in good agreement with the lowering of the surface observed from TanDEM-X satellite data (Fig. 4). However, upstream of the grounding line, the tip of the basal channel is located up to 400 m northwest of the center of the surface depression. Between 2018 and 2021, the channel grew, especially in the upstream area, where the channel height increased by almost 200 m (Fig. 5a, b). In contrast, no significant change in ice thickness nor channel height occurred downstream of the grounding line (Fig. 5c).

Airborne radar data from 1998, collected by Niels Reeh and Erik Lintz Christensen with the DTU (Technical University of Denmark) Space 60 MHz ice sounder, show no channel near the grounding line and a small 120 m high channel located 5 km downstream of the grounding line (Fig. 5d). The ice has also thinned considerably outside of the central channel. Along an across-ice-flow profile taken 600 m downstream of the grounding line, the average ice thickness in 2021 was 38 m less than in 2018 (Fig. 5c). Compared with 1998, the ice thickness 5 km downstream of the grounding line has decreased by more than 162 m (or 32 %). Above the large subglacial channel, the glacier has thinned by 67 %.

## 4.2 Extreme subglacial melting at the floating ice tongue

An analysis of the change in ice thickness at a given location (Eulerian perspective), as in the previous section, reveals changes in the geometry of the glacier. However, because the ice is flowing, considering the Lagrangian perspective in addition to the Eulerian is necessary for a full understanding of the process that causes these changes. The repeat UWB profile D–D′ from July 2021 is the Lagrangian repeat of the profile C–C′ from April 2018 (Fig. 1c). On average, the ice thickness at profile D–D′ in 2021 was reduced by 193 m, corresponding to a mean annual rate of 59 m a$^{-1}$. As the surface ablation is typically $< 2\,\mathrm{m\,a^{-1}}$ (Zeising et al., 2020) and the

dynamic thinning due to strain is low, as shown by all ApRES measurements (Fig. 3, Appendix B2), most of this thinning is attributed to basal melting.

In order to investigate the spatial distribution of Lagrangian thinning, we analyzed pRES measurements performed in July 2017 and 2018 at the same surface point. Figure 1b shows the spatial distribution of the thinning rates of all repeated pRES measurements (colored dots), while these are separated into nadir (colored dots with a line showing the flow path) and off-nadir thinning rates (colored area) in Fig. 1c. The marker shape of the off-nadir thinning rates in Fig. 1c corresponds to the scattering area from which the off-nadir basal reflections could have occurred. The thinning rates are between $1.7 \pm 0.1$ and $134 \pm 21\,\mathrm{m\,a^{-1}}$ for locations spread over the entire ice tongue of 79NG (Fig. 6). The highest (off-nadir) thinning rates of $126 \pm 20$ and $134 \pm 21\,\mathrm{m\,a^{-1}}$ were found at the most downstream bulge of the grounding line, next to the central subglacial channel where the ice draft is large. However, moderate thinning rates of $< 21\,\mathrm{m\,a^{-1}}$ were observed at a similar distance from the grounding line and for a similar draft (Fig. 6). Further downstream, although still within the hinge zone, we observed predominantly high thinning rates ($> 50\,\mathrm{m\,a^{-1}}$) spread across the entire width of the ice tongue. In general, thinning rates are observed to be below $30\,\mathrm{m\,a^{-1}}$ several kilometers downstream of the grounding line, declining towards the calving front to between $1.7 \pm 0.1$ and $3.2 \pm 0.1\,\mathrm{m\,a^{-1}}$ (Figs. 1, 6).

Variability on small spatial scales is accessible using a combination of nadir and off-nadir returns. At the pRES measurement location g4_p02 (Fig. 2), where we link the origin of the off-nadir reflection to a small subglacial channel, we derived two estimates of thinning rates: one is based on the repeated nadir reflection outside the channel ($6.4 \pm 0.1\,\mathrm{m\,a^{-1}}$), whereas the other is based on both first basal (off-nadir) reflections in 2017 and 2018 within the channel ($73 \pm 10\,\mathrm{m\,a^{-1}}$). This comparison indicates a growth of the subglacial channel by more than $66\,\mathrm{m\,a^{-1}}$.

The ApRES time series show a strong spatial and temporal variability in the basal melt rates without a clear seasonal cycle. All three ApRES devices recorded high melt rates of $> 50\,\mathrm{m\,a^{-1}}$ on average between October 2017 and July 2018; these rates decreased to $\sim 30\,\mathrm{m\,a^{-1}}$ until April 2019 and stayed low until the end of the record in January 2020 (ApRES2b), July 2022 (ApRES1), and September 2023 (ApRES4), respectively (Fig. 7). This change is particularly pronounced for ApRES1, which is located on the southeastern side of the glacier. Between April 2018 and April 2019, when ApRES1 was about 8–9 km downstream of the grounding line, the melt rate dropped from $137 \pm 2\,\mathrm{m\,a^{-1}}$ (95 % quantile) to just $30 \pm 1\,\mathrm{m\,a^{-1}}$ (Fig. 7a). After two periods with higher melt rates in the summer and autumn of 2020, the basal melt rate decreased to zero for almost all of the remaining 18 months. In early 2017, melt rates $> 120\,\mathrm{m\,a^{-1}}$ (95 % quantile) were recorded 5.5 km downstream of the grounding line on the northwestern side by

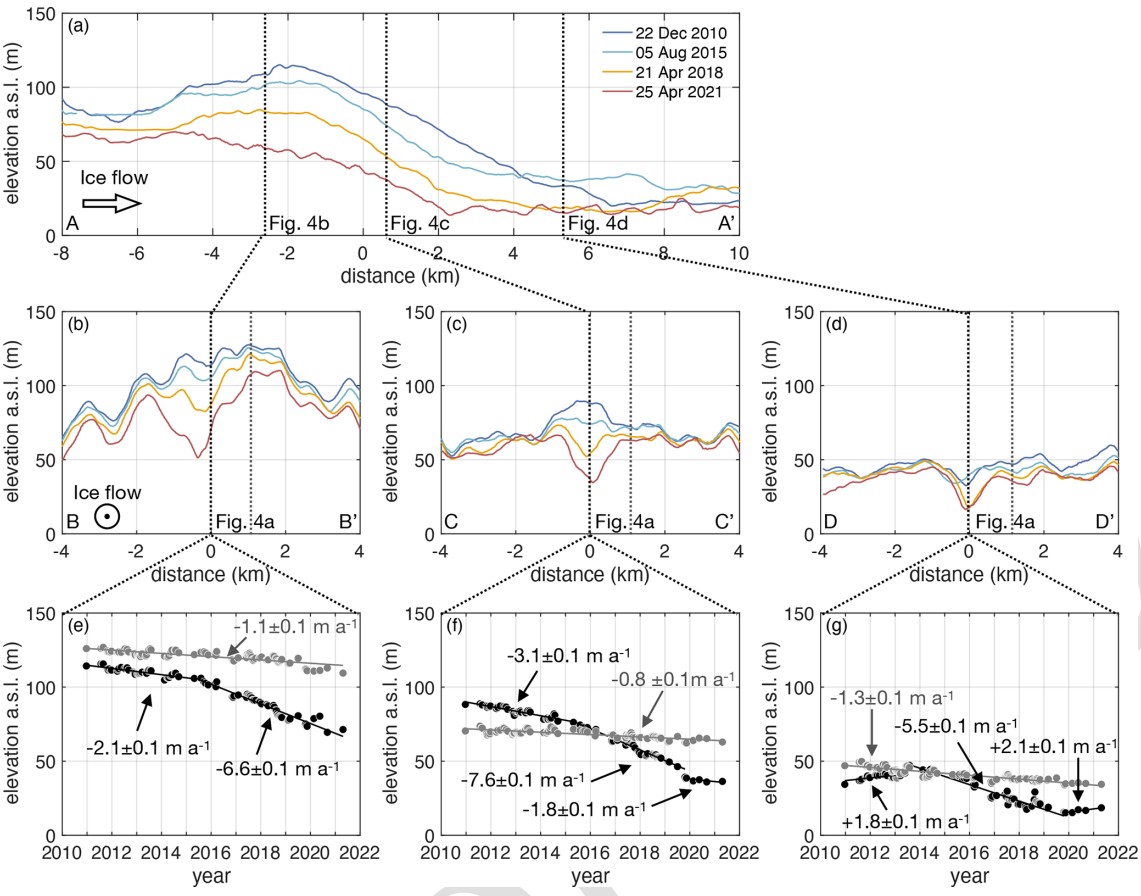

**Figure 4.** Surface elevation above sea level (EGM2008) from TanDEM-X satellite data between 2010 and 2021 **(a)** in the ice flow direction and **(b–d)** in the across-ice-flow direction. The distance in panel **(a)** is relative to the grounding line; in panels **(b)–(d)**, the distance is relative to the profile in panel **(a)**. The location of all profiles is shown in Fig. 1 and marked here using dashed lines. **(e–f)** Time series of surface elevation since 2010 at the three crossings above (black) and outside (gray) the graben-like structure. The numbers represent the gradient of the linear regression.

ApRES2a at the first basal return, while the concurrent median melt rate was below 50 m a$^{-1}$ (Fig. 7b). After the relocation of ApRES2a (now named ApRES2b) to its starting point in the summer of 2018, ApRES2b recorded a 50 m lower ice thickness and a 50 % lower melt rate (95 % quantile) than ApRES2a 2 years before. Furthermore, the spatial variability (difference between the median and the 95 % quantile) of ApRES2b was greatly reduced. The highest melt rates of 150–168 $\pm$ 5 m a$^{-1}$ lasting 17 d were recorded by ApRES3 at the beginning of the time series in July and August 2017. At that time, the ApRES3 device was located 3 km from the grounding line next to the large central basal channel. After these high melt rates dropped to roughly 50 m a$^{-1}$ following the summer in 2017, the basal melt rate generally showed a steady decrease to $\sim$ 20 m a$^{-1}$ until the end of the time series in September 2023.

## 5 Discussion

An analysis of the change in 79NG's geometry between 1998 (Reeh's airborne radar) and 2021 (this study) reveals a 32 % thinning in a narrow region 5 km from the grounding line as well as an ice base that has become channelized, especially in the vicinity of the grounding line (Fig. 5). Compared to the 30 % thinning observed by Mouginot et al. (2015) for the period from 1999 to 2014, the thinning has continued without accelerating. The onset of steep basal slopes has been shifted several kilometers in the upstream direction, especially within the large central channel (Fig. 5a). We associate this shift with enhanced basal melt rates that are above those required for a steady-state ice thickness, thus causing steep basal gradients. A remarkably similar change in geometry was found for a melt channel at Petermann Glacier between 2002 and 2010 (Münchow et al., 2014).

For the initialization of ice-sheet models, ice geometries, such as that from IceBridge BedMachine Greenland (Morlighem et al., 2017), are often used, which are based

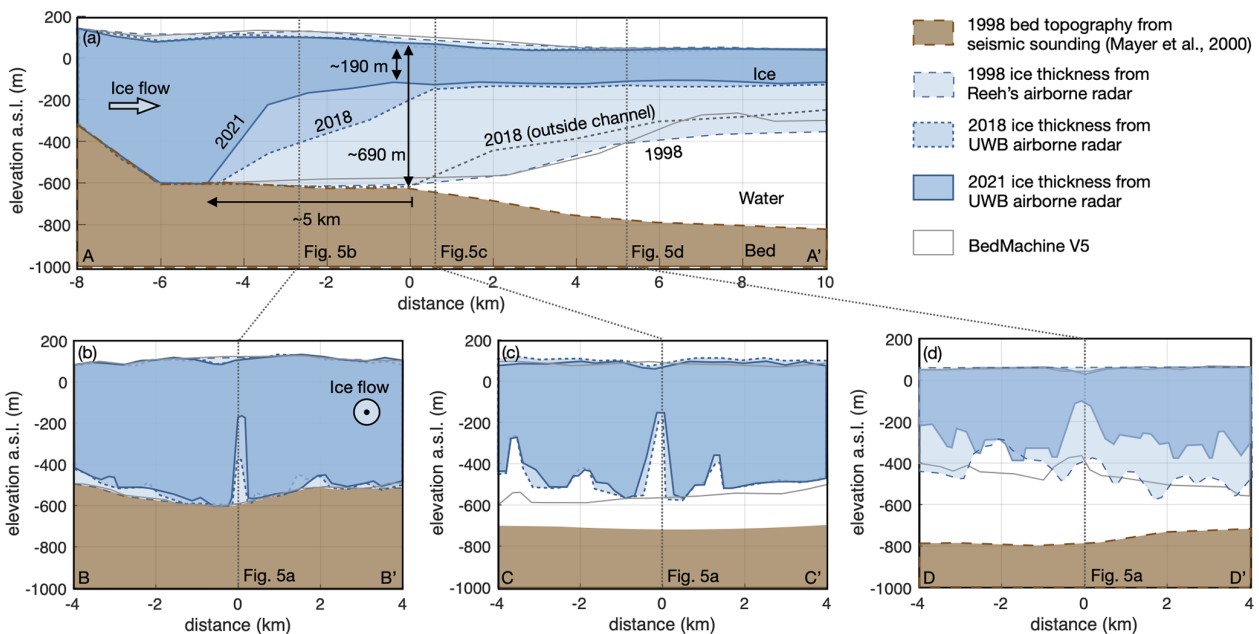

**Figure 5.** Ice thickness evolution of 79NG between 1998 and 2021 **(a)** in the ice flow direction and **(b–d)** in the across-ice-flow direction. The bed topography in panels **(a)** and **(d)** is based on active seismic measurements by Mayer et al. (2000) from 1998. The ice geometry in panels **(a)**–**(d)** is based on airborne radar measurements from 1998 (Reeh), 2018 (UWB), and/or 2021 (UWB). The IceBridge BedMachine Greenland, Version 5 (Morlighem et al., 2017, 2022) ice geometry is shown for the two across-ice-flow sections, panels **(c)** and **(d)**, of the floating part. The distance in panel **(a)** is relative to the grounding line; in panels **(b)**–**(d)**, the distance is relative to the profile in panel **(a)**. The locations of all profiles are shown in Fig. 1 and are marked here using dashed lines. Figure 4 shows the surface elevation change above these profiles. Appendix C shows the UWB data that this figure is based on.

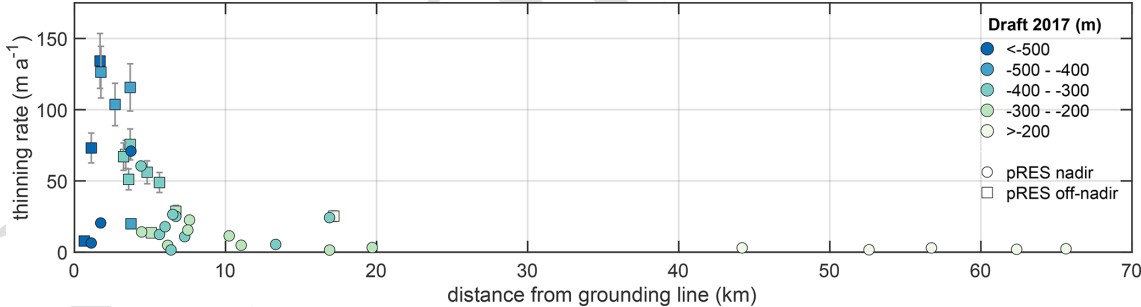

**Figure 6.** Distribution of the thinning rate as a function of distance from the grounding line. Color-coded draft of the floating tongue derived from pRES measurements and separated for nadir and off-nadir thinning rates. Most uncertainties are too small to visualize.

on a compilation of airborne radar measurements. At 79NG, ice thickness measurements since 1993 have been taken into account in IceBridge BedMachine Greenland, Version 5 (Morlighem et al., 2022), and the 2018 UWB data have also been included here. The comparison of the 2021 UWB ice thicknesses with BedMachine, Version 5, at the two across-ice-flow sections C–C′ and D–D′ shows differences of −91 ± 108 and −188 ± 56 m, respectively (Fig. 5c, d). This illustrates that the ice thickness of those glaciers, which have changed significantly in a few years due to the warming of the ocean and atmosphere, is difficult to represent. The impact of a more accurate, current ice thickness distribution

on the simulated evolution of floating ice tongues needs to be explored in regional studies such as that from Choi et al. (2017) for 79NG; however, this is beyond the scope of this study.

At Petermann Glacier, Washam et al. (2019) observed strong seasonal variations in the basal melt rates beneath the floating tongue, with summer melt rates being more than 4 times larger than in winter. In contrast to this, we see no evidence of seasonality in the melt rate time series for 79NG, despite the increase in melting at ApRES1 in July 2020. The absence of a summer increase in the basal melt rates is consistent with in situ measurements of ocean temperatures

 

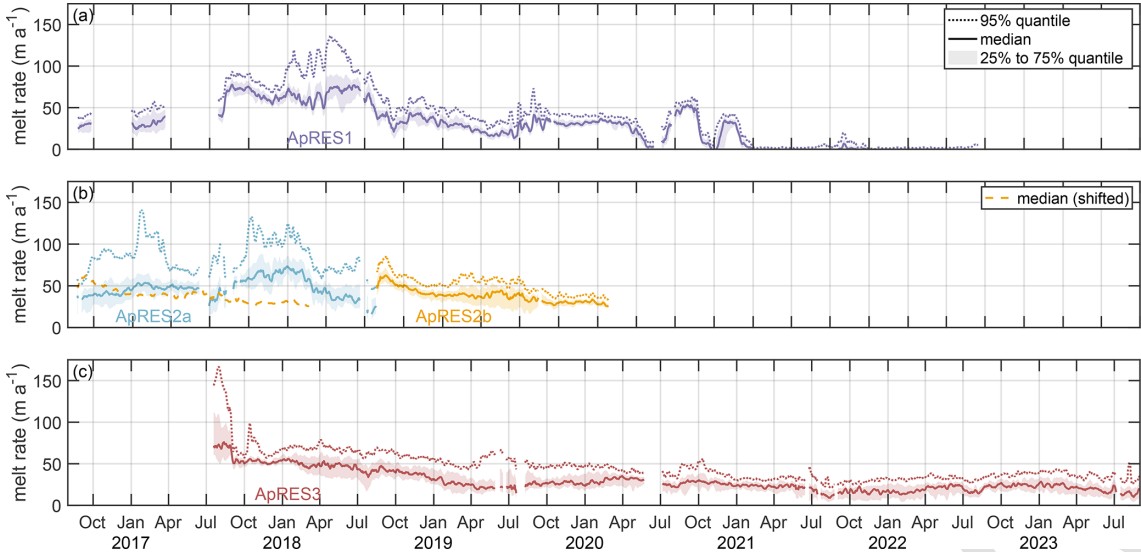

**Figure 7.** Basal melt rate time series of all ApRES measurements: **(a)** ApRES1, **(b)** ApRES2a and ApRES2b, and **(c)** ApRES3. The dashed line shows the 95 % quantile, the solid line shows the median, and the shaded area marks the range between the 25 % and 75 % quantiles. The dotted line in panel **(b)** represents the mean melt rate of ApRES2b shifted in time for a comparison with ApRES2a from an Eulerian perspective.

and velocities between September 2016 and September 2017 (Schaffer et al., 2020), showing persistent inflow of warm AIW into the cavity and an overlying outflow of glacially modified AIW throughout the year without a clear seasonal signal.

Combining the findings of this study with the observed inflow (Schaffer et al., 2020) and modeled (Reinert et al., 2023) currents below 79NG helps provide a full picture of the ice–ocean interaction at 79NG. Warm AIW flows over the sill into the cavity as a dense and saline bottom plume. As the keel of thick ice near the grounding line is exposed to this warm water, large amounts of heat are supplied to the ice base. The meltwater rises along the basal slope as a positively buoyant plume that may drive turbulent mixing with the warm AIW and, thus, intensify basal melting (Jenkins and Doake, 1991; Jenkins, 2011; Schaffer et al., 2020; Burchard et al., 2022).

To melt ice at a rate of $140\,\mathrm{m\,a^{-1}}$ requires a heat flux of between 1360 and $1600\,\mathrm{W\,m^{-2}}$ (see Appendix D) depending on the range of the glaciers' temperatures, which we assume to be between $\sim 0\,\mathrm{K}$ (temperate ice) and $30\,\mathrm{K}$ below the pressure melting point. This heat flux must be provided by the water in the cavity below 79NG. We assume a salinity of 34.5 psu (practical salinity units) and an ice draft of 320 m, estimated for the location of ApRES2a, where the highest melt rates of $140\,\mathrm{m\,a^{-1}}$ were determined during winter. Measurements of the inflow temperatures exceeded $1.2\,°\mathrm{C}$ at the calving front (Schaffer et al., 2020), corresponding to $2.9\,\mathrm{K}$ above the pressure melting point at the position of the observation. In order to produce a sufficiently high turbulent heat flux into the boundary layer for this given temperature,

an ambient velocity of $0.22\,\mathrm{m\,s^{-1}}$ is required for temperate ice and $0.26\,\mathrm{m\,s^{-1}}$ is needed for ice $30\,\mathrm{K}$ below the pressure melting point (see Appendix D). Previously simulated velocities of a buoyant plume rising along the ice base of 79NG indicate velocities of up to $0.22\,\mathrm{m\,s^{-1}}$ (Reinert et al., 2023). From these numbers, we conclude that the ocean currents underneath 79NG are able to supply a heat flux that is high enough to explain even the highest observed annual mean melt rates if they come into contact with the ice base.

The spread of thinning rates near the grounding line from near zero to $> 100\,\mathrm{m\,a^{-1}}$ may be related to the water column thickness distribution. A water column thickness of 50–140 m (Mayer et al., 2000) was found where we observe the highest basal melt rates and where the grounding line reaches farthest downstream. We do not have any information on water column thickness elsewhere. However, the southeastern part of the grounding line is situated on a mountainous landform. We hypothesize that only a shallow water column exists here, which prevents the flow of warm ocean currents toward the grounding line, resulting in the observed low thinning rates. Further downstream, the plume loses heat to the melting of ice and buoyancy via the entrainment of ambient water; thus, it cools down and eventually detaches from the ice base, leading to a strong decrease in basal melting for the thinner, more gently sloped areas of the floating ice tongue (Reinert et al., 2023). This concept is consistent with the low melt rates and glacially modified AIW observed at the calving front, where the outflowing water is $0.9\,\mathrm{K}$ cooler than the inflowing AIW (Schaffer et al., 2020).

While this picture accounts for the first-order, quasi-two-dimensional distribution of melt rates as well as the observed

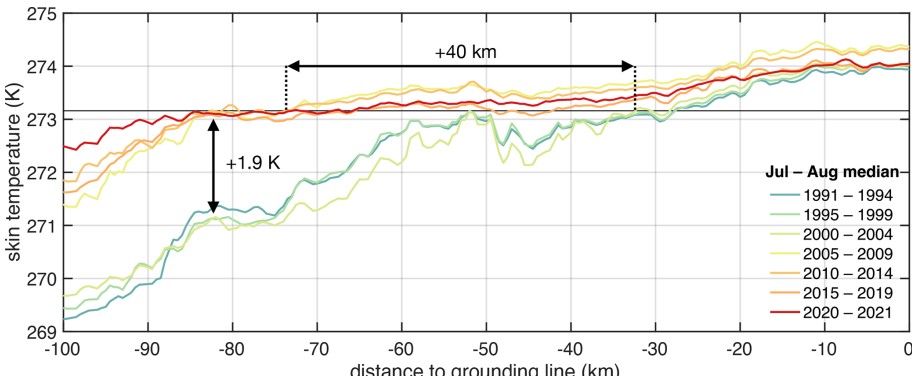

**Figure 8.** Development of skin temperature from the Copernicus Arctic Regional Reanalysis (Schyberg et al., 2020) along the central flow line since 1991. The temperature shown is the median skin temperature between 1 July and 31 August (at 15:00 UTC) for the given years.

hydrography, it does not explain the existence and growth of basal channels. In previous studies, the existence and location of basal channels have been linked to subglacial water discharge that rises along the basal slope inside a pre-existing basal channel and intensifies basal melting (Le Brocq et al., 2013; Marsh et al., 2016; Washam et al., 2019). We hypothesize that the same applies to the large basal channel at 79NG, where subglacial discharge might have caused the channel's growth in the upstream direction due to extreme basal melting. Unfortunately, we have no observations of nadir melt rates within a channel, as meltwater in summer accumulated in the surface depression above, preventing the deployment of an ApRES device. We analyzed the hydrostatic imbalance of the ice above the channel to assess the possibility of determining melt rates based on Lagrangian surface elevation changes. Therefore, we calculated the mean vertical ice density from the ice thickness and the surface elevation, recorded during the flight campaign in 2021 using the UWB airborne radar and laser scanner (see Appendix E). The result shows significantly lower densities of the ice above the channels, suggesting that the ice is not in hydrostatic equilibrium, which confirms the findings of Chartrand and Howat (2023). As this prevents the analysis of melt rates using satellite remote-sensing data, we can only draw conclusions from the basal geometry and its temporal changes using UWB airborne radar. High-resolution measurements of the basal topography at the Petermann and Thwaites glaciers using underwater vehicles have revealed steep-sided terraces and heterogeneous melting (Dutrieux et al., 2014; Schmidt et al., 2023). Because the UWB airborne radar does not allow us to resolve the base at a similar resolution, we consider only the average basal slope and, thus, interpret the average melt pattern.

At 79NG, the high melt rates occur primarily near the origin of the channel, where the greatest basal slope exists. With decreasing basal slopes inside the channel, the melt rate also decreases. This results in an upstream shift in the melt pattern compared with outside of the channel:

i. Upstream of the grounding line and downstream of the grounding line, where a low water column exists, higher melt rates occur inside the channel compared with outside of the channel.

ii. In the vicinity of the grounding line, where the ice is in contact with warm ocean currents, lower melt rates occur in the channel compared with outside of the channel. This pattern is consistent with observations from a basal channel at the Filchner Ice Shelf, Antarctica (Humbert et al., 2022). In addition, it explains the small-scale variability in melt rates that we observe at some pRES measurement locations (e.g., at g4_p02 in Figs. 1c and 2).

The steepening of an ice base indicates that the melt rates and the ice transport are not in equilibrium, which seems to have been the case at 79NG in the past when a warm AIW inflow was present. However, we have found indications of reduced heat transport into the cavity of 79NG since 2018. In that year, we observed a strong decrease in the melt rate at all ApRES sites that have remained low since. Additionally, the repeat of the ApRES2 measurements after 2 years shows that high melt rates between October 2016 and July 2018 have decreased the ice thickness at the starting location of the ApRES measurement by 50 m (Eulerian perspective). Due to the lower melt rates from July 2018 onward, the ice has thinned less than before (Lagrangian perspective). As a result, the ice thickness at the location where the measurement of ApRES2b stopped in December 2019 was even thicker than 2 years before.

Besides warm AIW inflow into the cavity, the melt rates can also be enhanced by an increase in subglacial discharge. Subglacial discharge has a seasonal component, including supraglacial lake drainage. The drainage of supraglacial lakes takes place on short timescales, even within only 1 d (Neckel et al., 2020). The lag between lake drainage and discharge across the grounding line is not well known, but it is reasonable to assume that the subglacial hydrological system is buffering water. Drainage of supraglacial lakes is

not restricted to the summer period, as the study of Schröder et al. (2020) also detected events in winter, which affects the timing of subglacial discharge. It is to be expected that subglacial discharge mainly affects the melt rates in the basal channels (Le Brocq et al., 2013). This appears to have been the case for 79NG over the past decade, as the central channel has grown and evolved in the upstream direction. In order to quantify if an increase in subglacial discharge has occurred, we roughly estimated the upstream extent of the average surface melt area from a simple analysis of the median summer skin temperatures (July and August) from the Copernicus Arctic Regional Reanalysis (Schyberg et al., 2020) along the ice flow line upstream of the large central channel (Fig. 8). This revealed a substantial increase in the median temperature (by up to 1.9 °C) since the year 2000. The area in which the temperature was above 0 °C on 50 % of the days was up to 32 km upstream during the 2000–2004 period but has extended to over 70 km since 2005–2009. Thus, intense warming of the atmosphere in the early 2000s increased the area of summer surface melt and most likely increased subglacial discharge. This is consistent with the finding at Petermann Glacier, where the subglacial water discharge has doubled since 2001 (Ciracì et al., 2023).

## 6 Conclusions

By combining geophysical in situ and remote-sensing methods, we revealed changes in the ice geometry of 79NG: the ice near the grounding line has become channelized and significantly thinner over the last 2 decades. Large, 500 m high subglacial channels originate several kilometers upstream of the grounding line. Here, higher melt rates occur inside the channel than outside, whereas we found evidence of higher melt rates outside the channels downstream of the grounding line. These high melt rates of $> 100\,\mathrm{m\,a^{-1}}$ are caused by thick ice that is in contact with the warm water masses at the bottom of the cavity. As these melt rates are above those required for a steady-state ice thickness, this leads to ice thinning from an Eulerian perspective and, thus, a steeper base slope. However, we also found low melt rates and small basal gradients under thick ice, particularly off the center of the glacier, which we attribute to a shallow water column thickness that prevents the flow of warm ocean currents toward the grounding line. As the ice thins in the downstream direction, the basal slope and the melt rates drop sharply, resulting in low values at the calving front. The temporal variation since September 2016 shows a non-seasonal variability and significantly decreasing Lagrangian melt rates with increasing distance from the grounding line. From 2018, these time series show a decrease in melt rates, suggesting a recent inflow of colder water into the cavity beneath the glacier. We conclude that warmer ocean inflow and increased subglacial discharge have caused the changes in ice geometry in the vicinity of the grounding line by forcing high basal melt rates. However,

based on our findings of thinning and upstream progression of subglacial channels, we cannot assess their impact on future stability. It would require numerical models as well as longer observational time series to evaluate the stability of 79NG and the Northeast Greenland Ice Stream; this should be addressed in further studies.

## Appendix A: Occurrence and identification of nadir and off-nadir reflection

We have identified different cases in which nearby basal channels affect the origin of the first recorded basal reflection in repeated pRES echograms (Table A1, Fig. A1). All have in common that the derived range differences between two measurements ($\Delta R$ derived) underestimate the nadir ice thickness ($\Delta H$ nadir) or the off-nadir ice thickness ($\Delta H$ off-nadir).

Several pRES echograms indicate the occurrence of numerous strong basal reflections (Fig. A2). For steep basal gradients, the off-nadir reflection may occur prior to the nadir reflection. We interpret the first basal reflection as an off-nadir reflection, as long as no further information reveals the true nadir reflection. The first off-nadir basal reflection in the first measurement and the repeated measurement can have occurred at two different locations (locations "A" and "B"). If this is the case, we can conclude that the melt rate has been higher at location B than at location A; otherwise, the first basal reflection would have occurred at location A in both measurements. However, when we compare the distance to A and to B, we know that the true change in ice thickness at B has been higher. Thus, we underestimate the thinning and the melt rate. If the second basal return occurred at an off-nadir angle, the estimated melt rate is below the vertical melt rate at that location. Still, the nadir melt rate can be even lower, but we cannot determine this melt rate.

**Table A1.** Possibilities regarding how basal channels affect the recording of nadir and off-nadir reflections. The notation used in the table is as follows: $t_1$ – time of first measurement; $t_2$ – time of repeated measurement; $H_1$ – ice thickness at $t_1$; $H_2$ – ice thickness at $t_2$; $\Delta H$ nadir – difference in ice thickness nadir; $\Delta H$ off-nadir – difference in ice thickness at off-nadir location; and $\Delta R$ derived – difference in depth at nadir projection.

| Case | Figure | $t_1$ | $t_2$ | $\Delta H$ |
|---|---|---|---|---|
| A | Fig. A1a | nadir | off-nadir | $\Delta H$ nadir $< \Delta R$ derived $< \Delta H$ off-nadir |
| | | Basal channel did not exist or was too small to be detected at $t_1$. At $t_2$, the growth of the channel is significantly larger than the ice thickness reduction nadir of the measurement device. | | |
| B | Fig. A1b | off-nadir | off-nadir | $\Delta R$ derived $< \Delta H$ off-nadir |
| | | Basal channel exists at $t_1$. At $t_2$, the ice thickness reduction nadir of the measurement device is not significantly larger than the growth of the channel. | | |
| C | Fig. A1c | off-nadir | nadir | $\Delta H$ off-nadir $< \Delta R$ derived $< \Delta H$ nadir |
| | | Basal channel exists at $t_1$. At $t_2$, the ice thickness reduction nadir of the measurement device is significantly larger than the growth of the channel. | | |
| D | Fig. A1d | off-nadir | off-nadir | $\Delta R$ derived $< \Delta H$ off-nadir |
| | | Two basal channel exist at $t_1$. At $t_2$, the ice thickness reduction nadir of the measurement device is not significantly larger than the growth of at least one of both channels. This type cannot be distinguished from Case B without known geometry. | | |

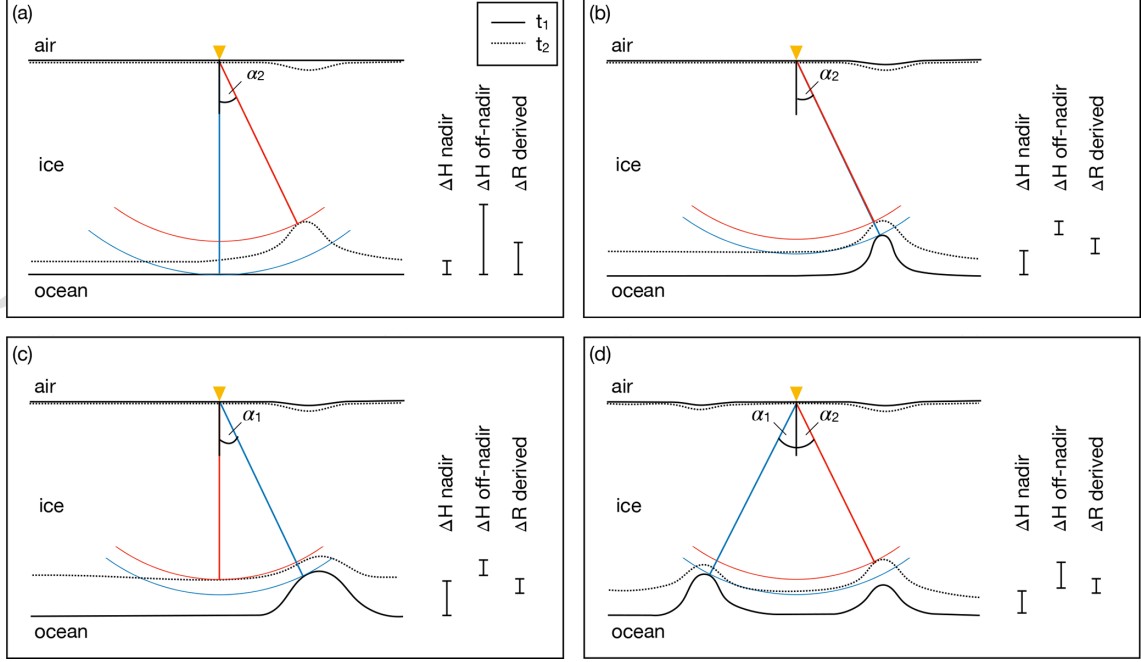

**Figure A1.** Sketch of off-nadir reflections and their influence on basal melt rates. The solid lines refer to the time of the first measurement (blue), $t_1$, and the dotted lines refer to the time of the repeat measurement (red), $t_2$. The yellow triangles mark the measurement positions. The red and blue straight lines mark the closest distance from the measurement to the ice base. The segments of a circle (up to 30° to nadir) correspond to the possible positions of the reflector with the shortest distance. The lengths of the bars on the right reflect the thinning of the ice between $t_1$ and $t_2$ for the position of the measurement ($\Delta H$ nadir), for the position of the closest reflector at $t_2$ ($\Delta H$ off-nadir), and for the range difference between the blue and red lines ($\Delta R$ derived). Note that at least one of $\Delta H$ nadir or $\Delta H$ off-nadir is always larger than $\Delta R$ derived.

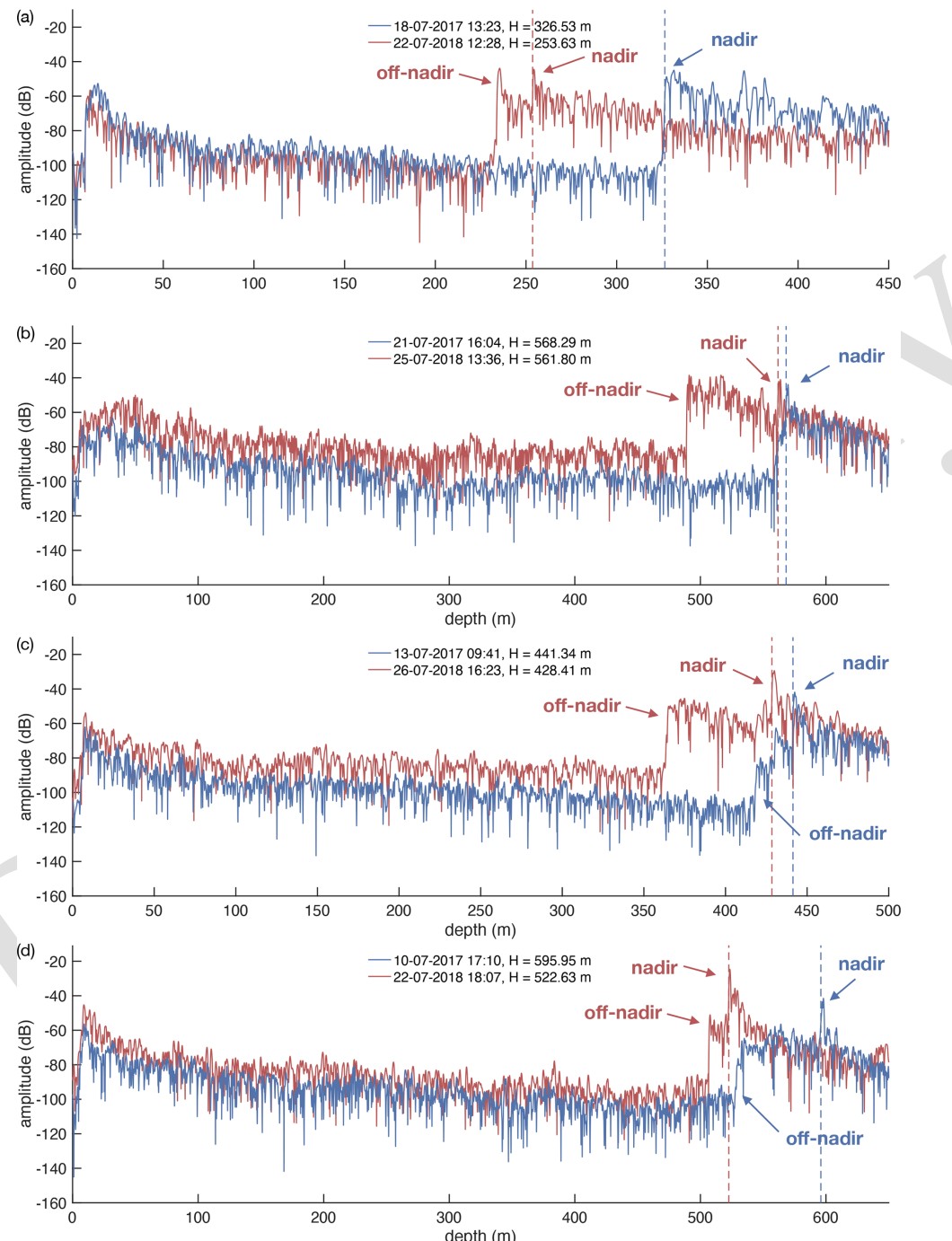

**Figure A2.** pRES measurements with the identified nadir and off-nadir reflections. Echograms from the first measurement are shown in blue, while those from the repeated measurement are shown in red. Vertical dashed lines mark the nadir basal return and, thus, represent the ice thickness $H$.

## Appendix B:  ApRES time series analysis

### B1    Estimation of ice deformation

Ice deformation affects the range $R$ from the ApRES to an englacial reflector located at $x_0$, $y_0$, and $z_0$ relative to the measurement location. ApRES measurements allow us to determine the depth profile of the vertical displacement of englacial reflectors relative to the surface and, thus, to compute the vertical strain $\varepsilon_{zz}$. The range displacement of an off-nadir reflector viewed at an angle $\alpha$ due to ice deformation $\Delta R_\varepsilon$ is also affected by the two horizontal normal $\varepsilon_{xx}$ and $\varepsilon_{yy}$ as well as the shear components $\varepsilon_{xz}$, $\varepsilon_{yz}$, $\varepsilon_{zx}$, and $\varepsilon_{zy}$. Accordingly, the location of a reflector shifts to $x_0 + \int_0^{x_0}(\varepsilon_{xx} + \varepsilon_{xz})\,\mathrm{d}x$, $y_0 + \int_0^{y_0}(\varepsilon_{yy} + \varepsilon_{yz})\,\mathrm{d}y$, and $z_0 + \int_0^{z_0}(\varepsilon_{zz} + \varepsilon_{zx} + \varepsilon_{zy})\,\mathrm{d}z$ at the time of a second measurement so that $\Delta R_\varepsilon$ can be calculated as follows:

$$
\Delta R_\varepsilon = \sqrt{
\begin{array}{l}
\left(x_0 + \int_0^{x_0}(\varepsilon_{xx} + \varepsilon_{xz})\,\mathrm{d}x\right)^2 \\[2mm]
+\left(y_0 + \int_0^{y_0}(\varepsilon_{yy} + \varepsilon_{yz})\,\mathrm{d}y\right)^2 \\[2mm]
+\left(z_0 + \int_0^{z_0}(\varepsilon_{zz} + \varepsilon_{zx} + \varepsilon_{zy})\,\mathrm{d}z\right)^2
\end{array}
}
\\
- \sqrt{x_0^2 + y_0^2 + z_0^2}. \tag{B1}
$$

For a nadir reflection ($\alpha = 0$) where $x_0 = 0$ and $y_0 = 0$, we assume that shear terms are negligible:

$$
\Delta R_\varepsilon^{\mathrm{n}} = \int_0^{z_0} \varepsilon_{zz}\,\mathrm{d}z, \tag{B2}
$$

where the range $R$ and, thus, $z_0$ equals the ice thickness $H$.

The estimation of $\Delta R_\varepsilon$ in the case of an off-nadir reflection requires the quantification of the normal and shear components as well as of $\alpha$ and $\beta$, which are unknown. In the following, we consider the shear terms to be small, as investigation of a melt channel on Filchner Ice Shelf (Humbert et al., 2022) has shown that the elastic shear strain is an order of magnitude lower than the strain in the normal direction. With channels appearing during our measurement period, the instantaneous elastic component is the one to be considered here. From the continuity equation (e.g., Cuffey and Paterson, 2010), we find that

$$
\varepsilon_{zz} = -\left(\varepsilon_{xx} + \varepsilon_{yy}\right), \tag{B3}
$$

which is the case for incompressible ice. Additionally, we know that $\alpha \leq 30°$ for the ApRES system so that the sum of the horizontal distances between the ApRES device and the reflector is smaller than or equal to the vertical distance: $x_0 + y_0 \leq z_0$. Thus, we can do the following quantification

$$
0 \leq |\Delta R_\varepsilon| \leq |\Delta R_\varepsilon^{\mathrm{n}}|, \tag{B4}
$$

where $\Delta R_\varepsilon$ and $\Delta R_\varepsilon^{\mathrm{n}}$ always have the same sign. This shows that strain thinning or thickening cannot be overestimated by assuming that a reflection occurred from a nadir scatterer.

## B2 ApRES echograms

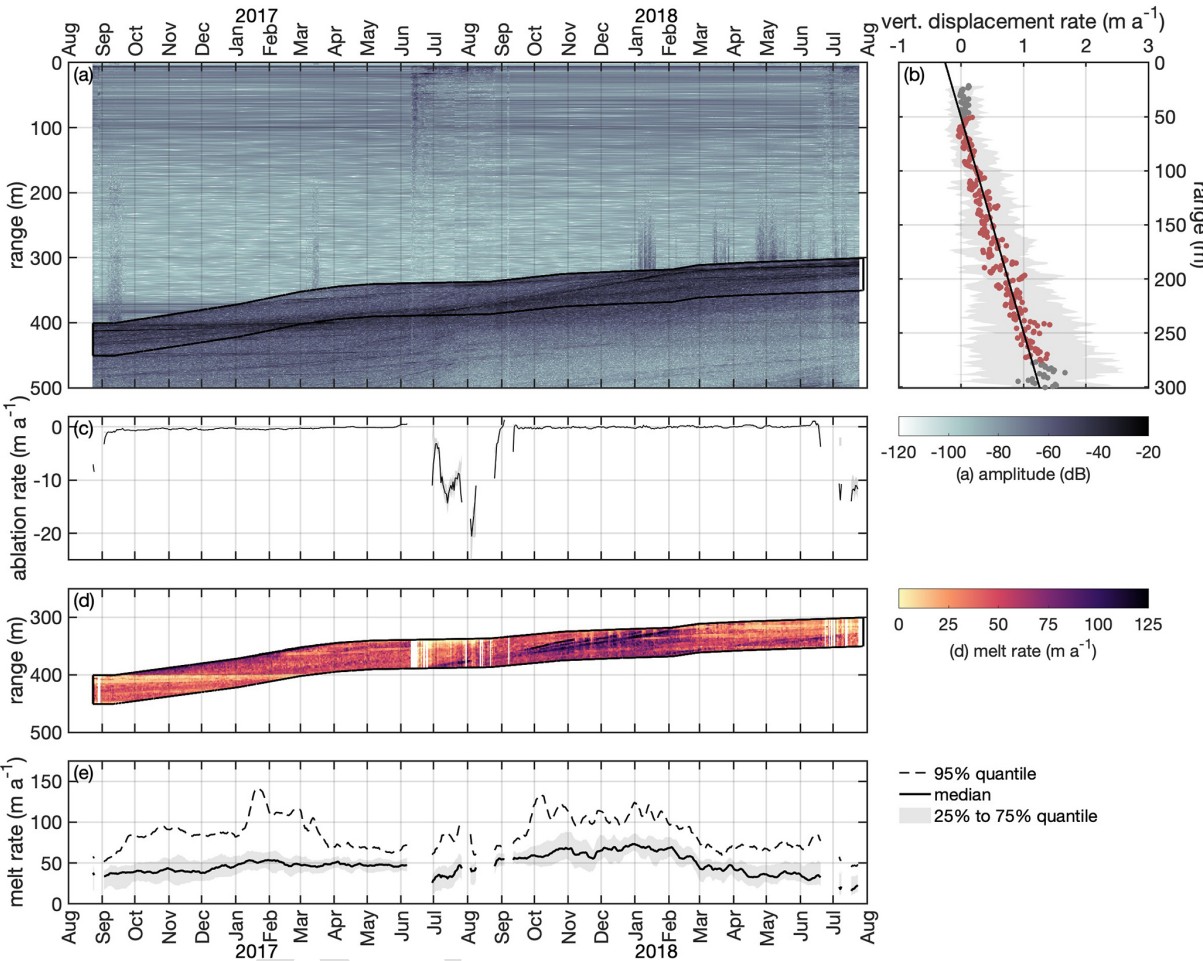

**Figure B1.** Analysis of the ApRES2a time series. **(a)** Time echogram of a Lagrangian measurement at ApRES2a recorded between August 2016 and July 2018. The black outline marks the first 50 m below the basal return. **(b)** Mean vertical displacement of englacial segments (dots). The gray shaded area marks the range between the 25 % and 75 % quantiles. Segments between 20 m below the surface and 20 m above the first basal return at the end of the measurement period (red dots) were used to calculate the change in ice thickness due to vertical strain by fitting a linear function (black line). **(c)** Time series of the ablation rate (negative for ablation). The gray shaded area marks the uncertainty due to the off-nadir correction. **(d)** Time series of the determined melt rate (color) within the first 50 m below the basal return, corresponding to the area marked by black lines in panel **(a)**. **(e)** Time series of the basal melt rate. The dashed line shows the 95 % quantile, the solid line shows the median, and the shaded area marks the range between the 25 % and 75 % quantiles.

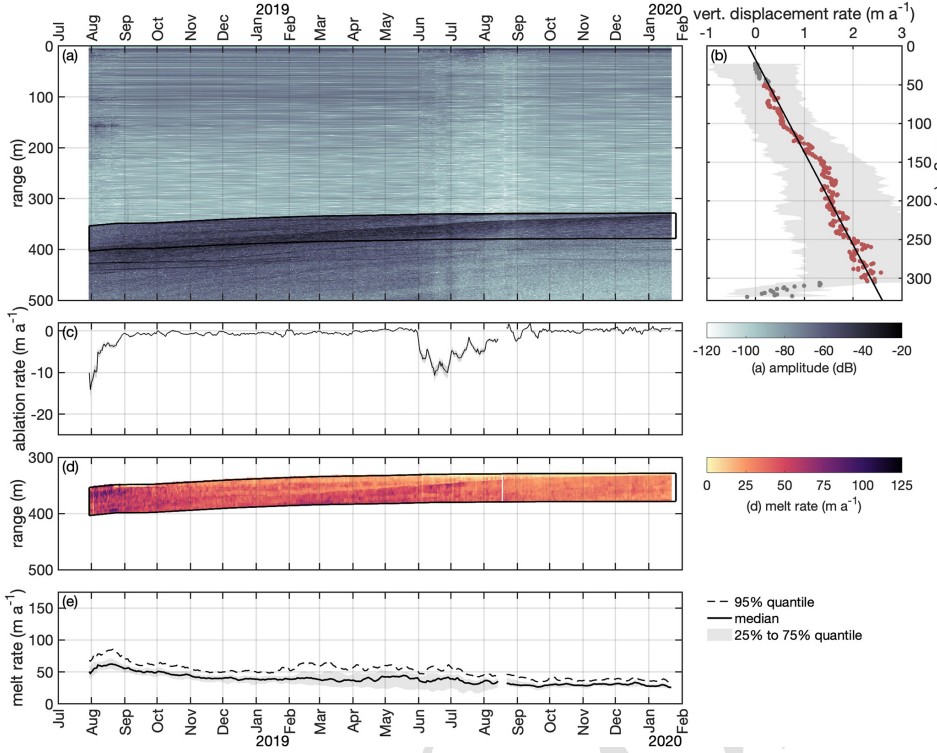

**Figure B2.** Same as Fig. B1 but for ApRES2b between July 2018 and January 2020.

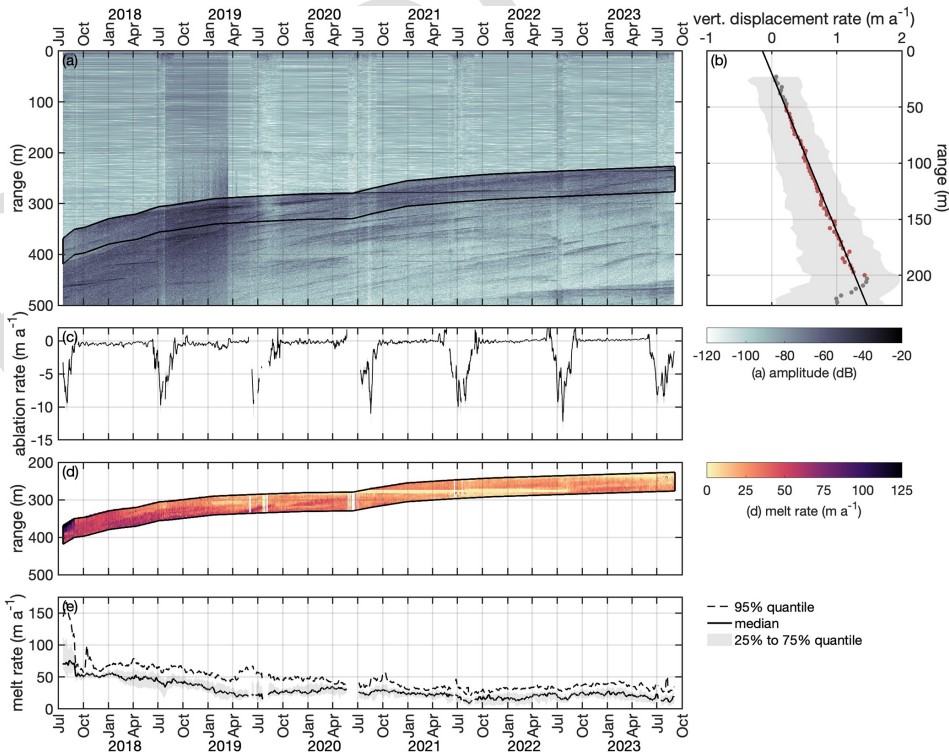

**Figure B3.** Same as Fig. B1 but for ApRES3 between July 2017 and September 2023.

## Appendix C: Subglacial channel observed by airborne radar

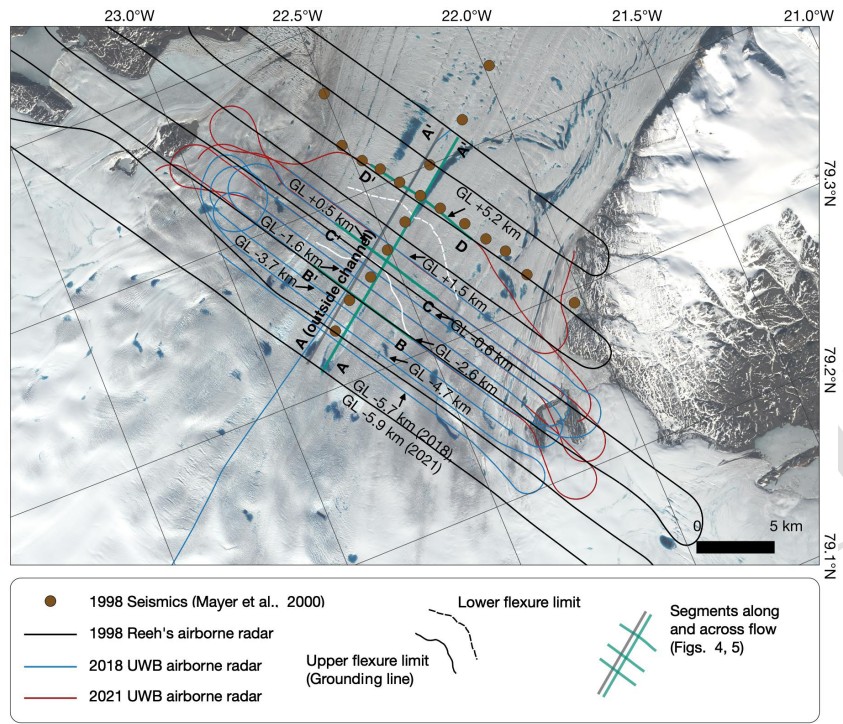

**Figure C1.** Map of seismic locations from 1998 (Mayer et al., 2000) and airborne radar data from 1998 (Reeh), 2018 (UWB), and 2021 (UWB). Copernicus Sentinel data from 2018 were retrieved from the Copernicus SciHub on 16 August 2021.

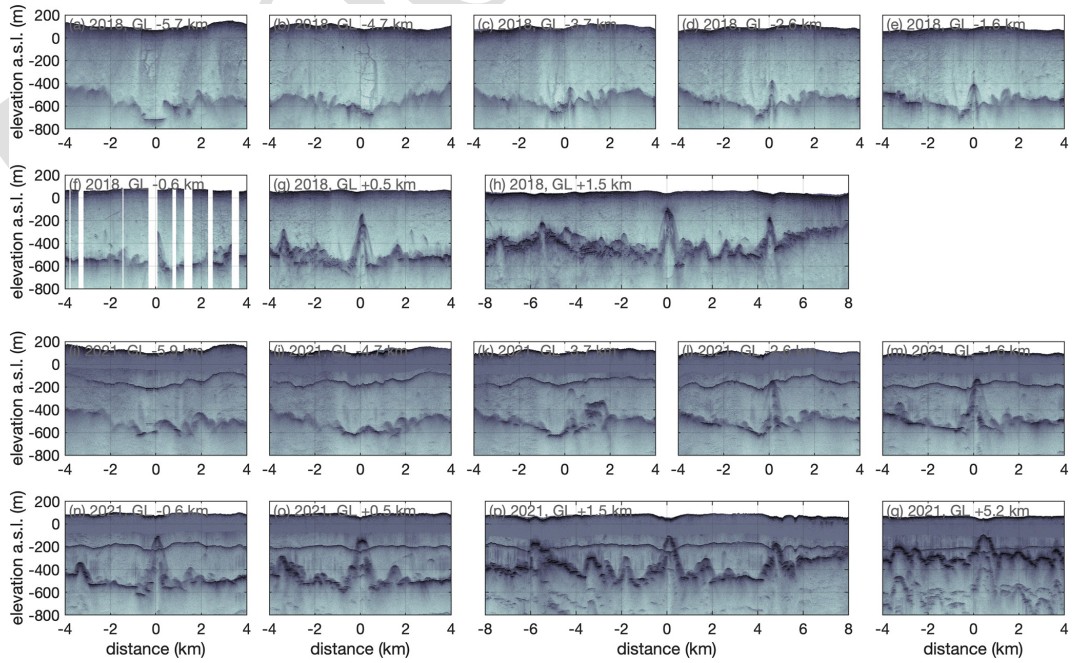

**Figure C2.** UWB airborne echograms across the basal channel **(a–h)** from 2018 and **(g–q)** from 2021. See Fig. C1 for the location.

## Appendix D: Oceanic heat flux

In order to estimate the oceanic heat flux $q_w$ required to sustain the basal melt rates $a_b$ (m s$^{-1}$) derived in this study, we separate the heat flux $q_w$ into two components: the heat flux to melt the ice ($q_m$) and the heat flux into the glacier interior ($q_i$) that is required for heating the ice by $\Delta T$ to the pressure melting point:

$$q_w = \underbrace{\rho_i \, a_b \, L}_{q_m} + \underbrace{\rho_i \, c_i(T) \, a_b \, \Delta T}_{q_i}. \tag{D1}$$

The heat fluxes depend on the density of the ice, $\rho_i = 917$ kg m$^{-3}$; the latent heat of fusion, $L = 334\,000$ J kg$^{-1}$; and the specific heat capacity for ice, $c_i(T) = 146.3 + 7.253 \cdot T$ [K] J kg$^{-1}$ K$^{-1}$ with the temperature $T$ in kelvin CE10 (Ritz, 1987).

To obtain an estimate of the heat flux that the ocean can provide, we follow the approach implemented in the Finite Element Sea ice–Ocean Model (FESOM; Timmermann et al., 2012). Here, a three-equation system is used that determines the temperature and salinity of a thin boundary layer along the ice-shelf base from its heat and freshwater exchange with the ice and the ambient ocean (Hellmer and Olbers, 1989; Holland and Jenkins, 1999). Besides the ocean temperature, the heat flux into this boundary layer is determined by the flow velocity in the ambient ocean, as the latter determines the friction and, thus, defines the turbulent fluxes of heat and salt (Jenkins and Doake, 1991).

## Appendix E: Hydrostatic imbalance near the grounding line

Basal melt rates can be estimated from surface elevation changes once hydrostatic flotation of the ice can be assumed. In the hinge zone, downstream of the upper flexure limit, bending dominates the vertical motion; hence, one has to assess the validity of the assumption of hydrostatic flotation. To validate this assumption near the grounding line of 79NG, we analyzed the hydrostatic imbalance by calculating the vertical mean ice density $\bar{\rho}_i$ assuming that the ice is in hydrostatic equilibrium. Here, we used the ice thickness $H$ from the UWB airborne radar data and surface elevation $h$ from the airborne laser scanner. Both data sets were obtained from the same flights in July 2021. The averaged vertical ice density can be calculated as follows:

$$\bar{\rho}_i = \rho_{oc} \frac{H - h}{H}, \tag{E1}$$

where $\rho_{oc} = 1028$ kg m$^{-3}$ is the density of the ocean. We defined a plausible range of vertical mean ice densities of between 900 and 917 kg m$^{-3}$. The results show high variability in the ice density in the hinge zone and also downstream where the ice is freely floating. Densities below 900 kg m$^{-3}$

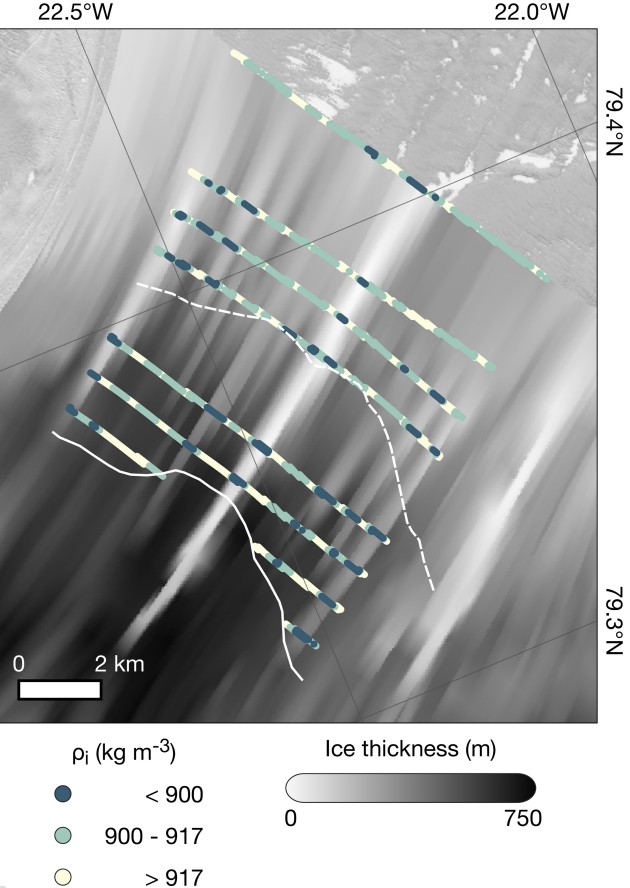

**Figure E1.** Computed ice density (dots) assuming the ice is in hydrostatic equilibrium based on ice thickness from UWB data (background) and surface elevation from airborne laser scanner data from the same flights in 2021. The solid white line shows the upper flexure limit (grounding line) and the dashed white line shows the lower flexure limit. CE11

(dark blue dots in Fig. E1) are reached above basal channels where the ice is thin, especially near the grounding line. This indicates that the ice above the channels near the grounding line is not in hydrostatic equilibrium, in contrast with the ice above smaller channels downstream of the hinge zone. Here, the ice density is widely above 900 kg m$^{-3}$, except above the large central channel. This result is consistent with the findings of Chartrand and Howat (2023). Near the grounding line, ice densities outside of the channels are above 900 kg m$^{-3}$ and widely also above 917 kg m$^{-3}$. The results show that hydrostatic equilibrium cannot be assumed in the hinge zone nor above the basal channels even further downstream, as one would expect from the viscoelastic material behavior of ice. Thus, in these areas, surface elevation changes cannot be used for the calculation of ice thickness changes.

*Data availability.* ApRES time series of basal melt rates (https://doi.org/10.1594/PANGAEA.928903; Zeising et al., 2024a), thinning rates derived from single repeated pRES measurements (https://doi.org/10.1594/PANGAEA.928541; Zeising et al., 2024b), and ice thickness data from the 2021 UWB survey (https://doi.org/10.1594/PANGAEA.963752; Zeising et al., 2023) are submitted to the World Data Center PANGAEA. Stake surface ablation and accumulation measurements from 2017 to 2018 (https://doi.org/10.1594/PANGAEA.922131; Zeising et al., 2020) are also available from the World Data Center PANGAEA.

*Author contributions.* OZ, DS, NN, and AH conducted the field expeditions, and AH and VH carried out the airborne campaigns. AH designed the study and planned the field expeditions and airborne campaigns. OZ processed the (A)pRES data and estimated and analyzed the resulting thinning and basal melt rates. ND processed the UWB 2018 data and discovered the central channel. VH processed the laser scanner data and the UWB 2021 data. NN processed all satellite data and determined the elevation changes and grounding line location. RT provided oceanographic expertise. OZ wrote the manuscript with contributions from all coauthors.

*Competing interests.* The contact author has declared that none of the authors has any competing interests.

ther geographical representation in this paper. While Copernicus Publications makes every effort to include appropriate place names, the final responsibility lies with the authors.

*Acknowledgements.* The pRES data were acquired as part of AWI's iGRIFF2017 and iGRIFF2018 campaigns. The airborne data were acquired as part of AWI's RESURV79 (2018) and 79NG-EC (2021) campaigns with AWI's polar aircraft (Wesche et al., 2016). The authors wish to thank Jens Köhler, Graham Niven, Tobias Binder, and Martin Gehrmann, the aircraft crews, Will Wilson, Dean Emberly, Stewart Clark, Marc-André Verner, Luke Cirtwill, Ryan Schrader, the team of Villum Research Station, the Arctic Command, and the 2018 and 2021 Station Nord teams for their support during the expeditions. TanDEM-X and TerraSAR-X data were made available through German Aerospace Center proposals GLAC7208 and HYD2059. Airborne DTU Space 60 MHz ice sounder raw data were processed into radargrams and ice surface as well as ice base data by Steen Savstrup Kristensen. The authors would like to thank Emerson E&P Software, Emerson Automation Solutions, for providing licenses within the scope of the Emerson Academic Program. We would like to acknowledge Steven Franke for his support with handling UWB data. Finally, we acknowledge Shfaqat Abbas Khan for servicing our ApRES during the lockdown in 2020 and retrieving our SD cards. CE12

*Financial support.* This research has been supported by the Bundesministerium für Bildung und Forschung (grant nos. 03F0778A and 03F0855A). Niklas Neckel has received funding from the European Union's Horizon 2020 Research and Innovation program under grant agreement no. 689443 within the framework of the iCUPE (Integrative and Comprehensive Understanding on Polar Environments) project.

The article processing charges for this open-access publication were covered by the Alfred-Wegener-Institut Helmholtz-Zentrum für Polar- und Meeresforschung.

*Review statement.* This paper was edited by Joseph MacGregor and reviewed by Christoph Mayer and one anonymous referee.

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

**Remarks from the language copy-editor**

CE1      Please note the change.

CE2      Please confirm the changes to this paragraph.

CE3      Please confirm the change.

CE4      Please confirm the change.

CE5      Please confirm the change.

CE6      Please note the change in all such instances.

CE7      Could this be changed to "single-iteration pRES measurements" in all instances? Based on your feedback, this may be clearer.

CE8      Please note the hyphen removal in all instances.

CE9      Please note the change (for clarity – "a" could be confused with a variable here).

CE10      Please note that, according to our house standards (and the ACS Style Guide p. 151), surnames that are used as units of measure are lowercase.

CE11      Please note that changes to values at this stage of the publication process require approval from the handling editor. Please provide a detailed explanation of why this needs to be changed. We will then start the post-review adjustment process and ask the handling editor for approval. Thanks.

CE12      Please confirm the slight changes.

**Remarks from the typesetter**

TS1      Please check adjustments made carefully once more.

TS2      In ranges and series, it is our house standard to retain only the final unit of measure. This is in line with the ACS Style Guide (p. 226). If this rule leaves room for misinterpretation in the context of your manuscript, the unit can be repeated. Please advise us on this matter.