# Peer review of "Extreme melting at Greenland's largest floating ice tongue"

_EGUsphere, 2023_

## Referee Comment (RC1)

**Review of egusphere-2023-1320**

*General comments:*

The manuscript "Extreme melting at Greenland's largest floating ice tongue" by Zeising et al. investigates melting beneath 79° North Glacier by synthesizing pRES, ApRES, airborne radar, and satellite SAR (TanDEM-X) measurements. They find channelized melt features and, indeed, extremely high melt rates, although the largest estimated melt rates (150 m/a) seem to be spatially localized. I found that the manuscript was exceptionally well-written with excellent figures, a clear and concise narrative, accessible description of phase-sensitive radar, and high scientific merit. In sum, I think that this is a great paper that could benefit from some more context, discussion, and comparisons with alternative methods. Below, I provide some specific comments and suggestions for further improving the manuscript that should be addressed prior to publication in *The Cryosphere*.

*Specific comments (major):*

1. **Introduction:** The introduction section is a little short as written, and I think could benefit from adding descriptions of the physics of channelized melting, how channelized features have also been found in Antarctica, methods for estimating the basal melt rate (e.g., explain more why you are using ApRES in the first place?), and perhaps any other ideas that arise in light of my other comments below. A good paper to reference on the observational side would be Alley et al. (2016), for example. (I see the description of channelization in the discussion, but some more in the introduction would be good too.)

2. **Comparison with surface-based estimation methods:** Clearly pRES is great for estimating basal melt rates. I do think though that somewhere you should further acknowledge the prevailing method for estimating basal melt rates, i.e., using satellite altimetry and surface velocity measurements under the assumption of hydrostatic (flotation) ice thickness. Ideally, since you have the elevation change, ice thickness, and ice surface velocity, you should be able to compare the estimates for either the melt rate or the true ice thickness vs. the hydrostatic ice thickness estimate. In particular, I would guess that your ApRES estimates are likely higher than hydrostatic-based estimates if the ice thickness is not perfectly hydrostatic around the channels due to deviatoric (bridging) stresses. This would be interesting in the context of recent modelling (Wearing et al., 2021) and observational (Chartrand & Howat, 2020,2023) studies that investigated the role of hydrostatic imbalance in surface-based melt-rate estimation; moreover, this would (A) highlight an advantage of ApRES in capturing internal strain rates that the hydrostatic methods do not include and (B) perhaps more directly relate the elevation-change measurements (or pRES thinning) to the ApRES melt rates in a conceptual sense. I think anything along these lines would be valuable/interesting to include given that you are near the grounding line and, thus, as you state in the introduction, the ice is probably not in "free flotation".

3. **Surface melting:** You suggest surface melting and the resulting enhanced subglacial discharge could cause enhanced melting. I think this could be improved in two ways. First, I think it would be good to generally discuss how surface hydrology and subglacial hydrology have been found to be linked at several of Greenland's outlet glaciers (e.g., Helheim Glacier), and that a subglacial outflow source for many ice-shelf channels has been hypothesized in Antarctica (e.g., Alley et al., 2016). Second, if there are any

indications of surface hydrology in this region in previous studies or satellite imagery you have looked at (e.g., Figure 1b?), that could be useful for further testing this hypothesis.

4. **Appendix D:** This Appendix is really only mentioned in passing in the discussion section, but describes some numerical calculations of ocean currents that are able to support the high melt rates. Consider including this material directly in a new results section (and/or the discussion) along with an explanatory/results figure if you are going to include it in the paper, which you absolutely should in my opinion if it helps explain the ApRES melt rates.

*Specific comments (minor):*
1. Line 5: I think you should include something about how the highest melt rates are spatially localized (i.e., later you say 95% quantile) and short duration here.
2. Line 30: "Bentley et al. (2023) gives evidence that the AIW…": suggest saying that this evidence comes from an epishelf lake.
3. Line 35: describe how meltwater alters fjord circulation (Straneo et al., 2016 ref)?
4. Line 105: Please clarify what "ice base – ice surface – ice base multiple" means
5. Equation 4: Define the vertical coordinate system somewhere, i.e., z is in (0,R), but what exactly do 0 and R mean?
6. Figure 1: For a while, I thought that there was a red star near ApRES2, but I see now that it is a black star with a red dot in it.  I think labelling the 2a and 2b endpoints on the map would help alleviate any confusion.
7. Line 185: "This can differ from the melt rate in the normal or vertical direction at the basal reflector." I got caught up on this statement, can you explain this in a little more detail? Related, in Appendix A you say "the resulting basal melting in the vicinity of the measurement is always underestimated, although the nadir melt rate might be lower", and I didn't completely understand that either.
8. Figure B1-B3: I think Including one of these in the main text would be good for understanding the ApRES data/method. I think plotting all of the components you use to calculate the melt rate ($\Delta R$, $\Delta R_s$, and $\Delta R_\varepsilon$) in panel c would be good, along with the melt rate you already have in panel d.
9. Equation (7): I don't entirely understand how you are calculating this in practice but I think the previous comment would help clarify.
10. Figure 4: I would remove the word "sketch" from the caption as it makes it sound like you are drawing something rather than plotting data
11. Figure 5: It is hard to see the BedMachine profile in this panel b (is it absent?). Also should probably include BedMachine citation in the caption
12. Line 225: Which figure are you referring to in Appendix B2 regarding small strains?
13. Line 230: "marker shape of the off-nadir thinning rates" add "in Figure 1" here to clarify
14. Fig 6a: Is there a negative melt rate/freezing towards the right or just zero?
15. In the discussion, I think some of the results concerning basal ice slopes could potentially be connected to some recent studies on the relation between basal ice slope (e.g., "terracing"; Dutrieux et al., 2014) and melt rates (Schmidt et al., 2023; Watkins et al., 2021). For example, on Line 205 you say "With decreasing basal slopes inside the channel, the melt rate also decreases", which is related to these ideas.

16. Line 337: I wasn't sure what you meant by "because they exceed such melt rates, which are necessary for a steady-state ice thickness"—I found this sentence confusing.
17. Line 338: "off the center"… center of the glacier? Suggest rewording
18. Appendix B1: On Line 370, what is $\beta$?
19. Equation B2: Are the shear terms neglected in the z integral in equation B1 to derive equation B2?
20. Appendix E: If you need to shorten the paper, I did not think this was strictly necessary.
21. Figure 7/Discussion: The surface temperature seems to drop slightly between 2005-2009 period and later years. Could this somehow be related to the decrease in melt rates? In general, more discussion of why the melt rates might be decreasing would be good. I know you say something about the "inflow of colder water", but could a diminishing subglacial outflow due to less surface melt also contribute?
22. Related to previous, you suggest a "recent inflow of colder water", just wondering if there are there any other observations available that might support this idea?
23. Table A1: In Case D, I was not sure what "simple measurements" meant
24. In the introduction, you talk about how basal melting may be related to ice shelf stability or disintegration. I think you should at least mention something about the stability of this system, and the uncertainties in that in the discussion. For example, do you think the channel is going to eventually break through the ice shelf thickness or otherwise destabilize the system somehow? Or, is it all very uncertain given the temporal dynamics of the melt-rate decreasing and possibly complex interactions with ice flow, ocean currents, and atmospheric changes?

*Technical corrections:*
a. Line 40: In the last sentence of the paragraph, I suggest reversing the order of clauses (i.e., "Other methods must be used to monitor…")
b. Line 165: Suggest changing "which results in an underestimated melt rate" to "underestimates the melt rate by X m/yr…" or similar. As written, I thought you meant that 2.7 m/yr was the absolute melt rate, not the underestimation amount.
c. Line 180: Change (Vaňková et al., 2021) to Vaňková et al. (2021)

**References:**

- Alley, K. E., Scambos, T. A., Siegfried, M. R., & Fricker, H. A. (2016). Impacts of warm water on Antarctic ice shelf stability through basal channel formation. *Nature Geoscience*, 9(4), 290-293.
- Chartrand, A. M., & Howat, I. M. (2020). Basal channel evolution on the Getz Ice Shelf, West Antarctica. *Journal of Geophysical Research: Earth Surface*, 125(9), e2019JF005293.
- Chartrand, A. M., & Howat, I. M. (2023). A comparison of contemporaneous airborne altimetry and ice-thickness measurements of Antarctic ice shelves. *Journal of Glaciology*, 1-14.
- Dutrieux, P., C. Stewart, A. Jenkins, K. W. Nicholls, H. F. J. Corr, E. Rignot, and K. Steffen (2014), Basal terraces on melting ice shelves, Geophys. Res. Lett., 41, 5506–5513, doi:10.1002/ 2014GL060618.
- Schmidt, B. E., Washam, P., Davis, P. E., Nicholls, K. W., Holland, D. M., Lawrence, J. D., ... & Makinson, K. (2023). Heterogeneous melting near the Thwaites Glacier grounding line. Nature, 614(7948), 471-478.
- Watkins, R. H., Bassis, J. N., & Thouless, M. D. (2021). Roughness of ice shelves is correlated with basal melt rates. Geophysical Research Letters, 48(21), e2021GL094743.
- Wearing, M. G., Stevens, L. A., Dutrieux, P., & Kingslake, J. (2021). Ice-shelf basal melt channels stabilized by secondary flow. *Geophysical Research Letters*, 48(21), e2021GL094872.

---

## Referee Comment (RC2)

Review of

**Extreme melting at Greenland's largest floating ice tongue**

by Zeising et al., submitted to *The Cryosphere*

The manuscript presents an analysis of a combination of measurements for detecting the local balance conditions in the grounding line region of Nioghalvfjerdsfjorden Glacier (79NG) in NE-Greenland. For this purpose the authors use pRES and ApRES and ultra-wide-band airborne radar measurements, complemented by surface elevation models, derived from TanDEM-X imagery. The geometry data are also compared to earlier seismic investigations form 1998, in order to analyse the long-term changes of the grounding zone of 79NG. They find extremely high melt rates on a local scale, but still considerable strong subglacial melting across the entire grounding zone.

The manuscript is clear and well written and presents a detailed analysis of data quality and comparison of data from different sources. Data and results are very well presented. In general, the results are based on a rigorous processing and analysis approach and provide new insight into the recent and medium term evolution of the grounding zone of 79NG. The manuscript will add important new knowledge to the scientific efforts of understanding the complex interaction of ice, ocean and climate in NE Greenland.

Apart from some minor issues, which I list further down, there is only one major question concerning the localised detection of incised channels into the underside of the glacier. The strong increase of the channel height is documented by UWB radar data between 2018 and 2021 and on a longer time scale by pronounced and locally concentrated surface lowering from SAR imagery. It was stated the measurements of Mayer et al. (2000) show no indication of subgacial channels close to the grounding line in 1998. However, the seismic measurements were performed with a 24 channel instrument, covering horizontal distances of 240 m. The single measurements were up to 2 kilometres apart and the final figure in Mayer et al. (2000) only shows an interpolated cross profile of the single shots. Therefore, it cannot be concluded that there was an absence of subglacial channels in 1998. There exists an unpublished data set of airborne radar data also from 1998 (named the "Niels Reeh data set" in e.g. Seroussi et al., 2011) which shows a much more detailed ice bottom topography in the grounding line region of 79NG. The figure shows a cross profile in the vicinity of the BB′ profile, with large subglacial undulations across the entire glacier, where the deepest reaches more than 200 m.

[Figure]

Fig.: RES cross profile in the grounding line region of 79NG from 1998. The dark blue line shows the profile location in Lat/Lon. The orange and bright blue lines show the ice geometry, while the yellow dot indicates the location of the subglacial channel identified in the recent manuscript.

Therefore, I highly recommend to consult to airborne RES data from 1998, in order to reach to sensible conclusions with regard to the temporal evolution of localised melt features.

Minor comments:

L. 51: I might be useful to already mention the data source of the DEMs here (e.g. from InSAR processing)

L 52: GROCE needs a reference.

L. 53: either "an UWB radar", or the "UWB radar" and adding some information.

L. 68: the spatial adjustment requires some error estimate.

L. 80: I do not understand this sentence. What do you presume in the hardware?

L. 85: The processing steps change the ground resolution, which should be discussed here.

L. 90: Accuracy of the laser measurements?

L. 105: The off-nadir reflections depend on the location of the instrument. This could be demonstrated in more detail here.

L. 135: This is unclear to me. Does that apply generally to single-repeat pRES measurements, or is this a special case?

L. 167: You state that you are able to estimate the $\Delta R^n$, why are you underestimating the melt rate then?

L. 180: citation format needs change

L. 188: The inaccuracy of signal propagation speed does not depend on the melt rate. I you would like to state that the uncertainties in the propagation speed result in similar inaccuracies as about 1% of the melt rate do, this should be reformulated.

L. 195/196: To which width does the region of surface lowering reduce in which distance?

L. 201: In which distance free floating occurs and what are the criteria for "free flotation"?

L. 222: The Lagrangian perspective also tells only one side of the story. Only the combination completes the information.

L. 222: I would be good state again that the profiles are taken from Fig. 1

L. 260: The 42% thinning are restricted to a narrow region, compared to the 79NG total extent. This should be mentioned here.

L. 270/271: This is true for significant changes in general, but applies also for warmer termperatures and therefore enhanced melt rates.

L. 281: As long as the pinning points exist at the front, seasonal changes cannot be expected.

L. 307/308: The low melt rates upstream of the grounding line and outside the large channels depend on what? Is there a patchy grounding line, or do you expect a distributed drainage system, if you refer to low water columns?

Seroussi, H., Morlighem, M., Rignot, E., Larour, E., Aubry, D., Ben Dhia, H., & Kristensen, S. S. (2011). Ice flux divergence anomalies on 79north Glacier, Greenland. Geophysical Research Letters, 38(9).

---

## Author Response (AR1)

**Authors response on Editor Comment #1 to egusphere-2023-1320**

Dear Editor, dear Joe MacGregor,

We are grateful for the two reviews we received, which helped to improve the quality of the manuscript. We followed the major comments from Reviewer 1 and extended the introduction by two sections about basal melt channels and ApRES. We analyzed the hydrostatic imbalance of the ice above the channel, added a few sentences about surface melting, and moved a part of the Appendix D section to the discussion. Reviewer 2 recommended using the unpublished Reeh's airborne radar data from 1998 to analyze the existence of the channel instead of seismic data from the same year. We got access to this data set and updated the text as well as Figs. 5 and C1. We have been informed that this data set will be published soon. Thus, we have not included a data set citation in this version of the manuscript, but we will add one as soon as the data set is published.

We have made additional changes that were not mentioned in the responses to the reviewer:

- We updated Fig. 1 with the correct thinning rates. Unfortunately, the thinning rates shown in Fig. 1 were not the most recent in the submitted versions. This results in a slight shift in the thinning rate at a few stations. The method, the values mentioned in the text, and those in Figure 5 were all correct.
- We became aware of a publication by Burchard et al. (2022), that we added in the discussion, without changing or adding a sentence.
- We changed the reference of Reinert et al. (2023) from the preprint to the published article.
- We added citations for data sets in the "Data availability" statement with temporally doi's.
- We corrected a typo: The required ambient velocity of the plume to produce a heat flux of 1600 W/m^2 is 0.26 m/a, not 0.27 m/a.
- We extended the time series of ApRES3 with new data that we collected this summer and updated Figs. 7 and B3. The extended time series strengthens the results but it does not change or add any new results.
- We added a threshold in the quantification of the ablation to remove outliers: "We removed outliers defined by ablation rates $> 0.1$ m d$^{-1}$"
- Reviewer 1 suggested removing Appendix E ("Surface skin temperature") if the manuscript needs to be shortened, as Appendix E is not essential for the manuscript. We agree to this point and would leave the decision to the Editor.

Many thanks for your efforts to improve our manuscript!

Best wishes
Ole and co-authors

**Authors point-to-point responses Referee Comment #1 to egusphere-2023-1320**

Please find the author's responses in blue below the reviewer's comments.
Please find the implemented changes in red below the responses.

Review of egusphere-2023-1320

*General comments:*
The manuscript "Extreme melting at Greenland's largest floating ice tongue" by Zeising et al. investigates melting beneath 79° North Glacier by synthesizing pRES, ApRES, airborne radar, and satellite SAR (TanDEM-X) measurements. They find channelized melt features and, indeed, extremely high melt rates, although the largest estimated melt rates (150 m/a) seem to be spatially localized. I found that the manuscript was exceptionally well-written with excellent figures, a clear and concise narrative, accessible description of phase-sensitive radar, and high scientific merit. In sum, I think that this is a great paper that could benefit from some more context, discussion, and comparisons with alternative methods. Below, I provide some specific comments and suggestions for further improving the manuscript that should be addressed prior to publication in *The Cryosphere*.

*Specific comments (major):*

1. **Introduction:** The introduction section is a little short as written, and I think could benefit from adding descriptions of the physics of channelized melting, how channelized features have also been found in Antarctica, methods for estimating the basal melt rate (e.g., explain more why you are using ApRES in the first place?), and perhaps any other ideas that arise in light of my other comments below. A good paper to reference on the observational side would be Alley et al. (2016), for example. (I see the description of channelization in the discussion, but some more in the introduction would be good too.)

   We agree that the paper would benefit from a description of the formation of basal channels and observations of channelized melting. When we were writing the original manuscript we somehow did not have in mind to get in the introduction already in the topic of the channels, but it is a very good idea and we are more than happy to include this. Thus, we can introduce the ApRES already and mention its advantages. We will add both to the introduction of the revised version.

   We added the following section to the introduction (Line 41 – 56):
   *"Subglacial water discharge from beneath the grounded ice is often linked to the location of basal channels in the floating ice shelves caused by locally enhanced melting (Le Brocq et al., 2013). Such channels can be up to a few kilometers in width and up to a few hundred meters in height (Rignot and Steffen, 2008). The spatial distribution of basal melt rates can be investigated using repeated measurements with the phase-sensitive Radio Echo Sounder (pRES). The same device can be operated in an autonomous mode (henceforth ApRES) to perform measurements over a longer period of time with a defined interval. Previous studies used pRES and ApRES measurements to investigate the spatial distribution and temporal variability of basal melting*

*inside basal channels: At the Ross Ice Shelf, Antarctica, Marsh et al. (2016) found enhanced melting inside a channel near the grounding line which reduced in the downstream direction. Humbert et al. (2022) revealed for a channel at Filchner Ice Shelf, Antarctica that melt rates inside the channel decrease in the direction of ice flow and fall below those outside the channel, causing the channel height to decrease. While Humbert et al. (2022) found no pronounced seasonality of melting inside the channel, Washam et al. (2019) detected a significant increase in melting inside a channel at Petermann Gletscher, Greenland during the surface melt period in summer. They linked the seasonality to the increased subglacial discharge that enhanced the inflow of warmer ocean currents into the cavity (Shroyer et al., 2017; Washam et al., 2019). Whether basal channels stabilize or weaken shelf ice is not fully understood yet (Alley et al., 2016). Numerical models indicate that the existence of channels can decrease the mean basal melt rate (Millgate et al., 2013), at the same time, the channels can structurally weaken the ice shelf (Vaughan et al., 2012)."*

2. **Comparison with surface-based estimation methods:** Clearly pRES is great for estimating basal melt rates. I do think though that somewhere you should further acknowledge the prevailing method for estimating basal melt rates, i.e., using satellite altimetry and surface velocity measurements under the assumption of hydrostatic (flotation) ice thickness. Ideally, since you have the elevation change, ice thickness, and ice surface velocity, you should be able to compare the estimates for either the melt rate or the true ice thickness vs. the hydrostatic ice thickness estimate. In particular, I would guess that your ApRES estimates are likely higher than hydrostatic-based estimates if the ice thickness is not perfectly hydrostatic around the channels due to deviatoric (bridging) stresses. This would be interesting in the context of recent modelling (Wearing et al., 2021) and observational (Chartrand & Howat, 2020,2023) studies that investigated the role of hydrostatic imbalance in surface-based melt-rate estimation; moreover, this would (A) highlight an advantage of ApRES in capturing internal strain rates that the hydrostatic methods do not include and (B) perhaps more directly relate the elevation-change measurements (or pRES thinning) to the ApRES melt rates in a conceptual sense. I think anything along these lines would be valuable/interesting to include given that you are near the grounding line and, thus, as you state in the introduction, the ice is probably not in "free flotation".

We understand that there is a need to compare in-situ observations of e.g. ice thickness and melt rates with surface-based estimates from remote sensing. However, we do not see this as the focus of our study but rather make our data set available for future remote sensing studies to validate their products. A comparison of (A)pRES-derived ice thicknesses and melt rates of this study with satellite-remote sensing-derived products is challenging. Ice thicknesses could only be compared where we observed the nadir ice thickness with (A)pRES measurements. Between the upper and lower flexure limit, there are only two to three sites where we identified the nadir ice thickness. Since we have no measurements inside a channel, we cannot compare the ice thickness

above channels based on (A)pRES measurements. However, we may be able to compare the melt rates. To compare the melt rate with the surface elevation time series from TanDEM-X, we have to calculate a Lagrangian dh/dt for the period of (A)pRES observations and correct these for tides and ice deformation from a velocity data set. The resulting uncertainty of such a melt rate might be too large to investigate if the melt rate estimate between the upper and lower flexure limit differs from the ApRES results due to the hydrostatic imbalance.

However, there might be another possibility to investigate the hydrostatic imbalance by comparing the airborne radar-derived ice thickness with the ice thickness estimated based on the surface elevation product from the simultaneously acquired laser scanner data.

We will test both possibilities. Depending on whether a reliable statement can be made, we may include this in the revised manuscript.

We have analyzed the hydrostatic imbalance by calculating the ice density $\rho_i$ assuming that the ice is in hydrostatic equilibrium. Here, we have used the ice thickness $H$ from UWB radar data and surface elevation $h$ from the airborne laser scanner. Both data sets were measured at the same flights in 2021. The averaged vertical ice density can be calculated as follows:

$$\rho_i = \rho_{oc} \frac{H-h}{H},$$

where $\rho_{oc} = 1028$ kg/m$^3$ is the density of the ocean.
We defined a plausible range of vertical mean ice densities between 900 and 917 kg/m$^3$.

The results show high variability of the ice density, in the hinge zone and downstream where the ice is freely floating.
Densities below 900 kg/m$^3$ (dark blue dots in Fig. 1a) are reached above basal channels where the ice is thin, especially near the grounding line. This indicates that the ice above the channels near the grounding line is not in hydrostatic equilibrium which is the case for the smaller channels downstream of the hinge zone. Here, the ice density is widely above 900 kg/m$^3$, except above the large central channel. This result is consistent with the findings from Chartrand and Howat (2023). Near the grounding line, ice densities outside of the channels are above 900 kg/m$^3$ and widely also above 917 kg/m$^3$.

The results show that the hydrostatic equilibrium can not be assumed in the hinge zone and not above basal channels even further downstream. Thus, in these areas, surface elevation changes can not be used for the calculation of ice thickness changes.
We have calculated thinning rates from a Lagrangian TanDEM-X to compare them with pRES-derived ice thickness changes. We found a strong scattering resulting in a standard deviation of 20 m/a, although the mean difference was close to zero. It is worth mentioning that our pRES study focuses on the area

near the grounding line (<10 km distance) which may significantly influence the scattering.

[Figure]

Fig. 1: (a) Computed ice density (dots) assuming the ice is in hydrostatic equilibrium based on ice thickness from UWB data (background) and surface elevation from airborne laser scanner data recorded on the same flights in 2021. (b) Lagrangian thinning rates from TanDEM-X (background) for the time period 20 July 2017 – 09 Aug 2018. The pRES-derived thinning rates are shown by colored dots (nadir) and by scattering areas (off-nadir), measured between July 2017 and July 2018.

We added a sentence in the introduction (Line 59 – 61):
"Particularly within a few kilometers from the grounding line and above basal channels, the ice is in hydrostatic imbalance, limiting the analysis of melt rates based on changes in surface elevation (Chartrand and Howat, 2023)."

Additionally, we added/modified a paragraph in the discussion (Line 354 – 361):
"Unfortunately, we have no observations of nadir melt rates within a channel since meltwater in summer accumulated in the surface depression above, preventing the deployment of an ApRES. We analyzed the hydrostatic imbalance of the ice above the channel to assess the possibility of determining melt rates based on Lagrangian surface elevation changes. Therefore, we calculated the mean vertical ice density from the ice thickness and the surface elevation, recorded during the flight campaign in 2021 using the UWB airborne radar and laser scanner. The result showed significantly lower densities of the ice above the channels, suggesting that the ice is not in hydrostatic equilibrium, which confirms the findings of Chartrand and Howat

(2023). Since this prevents the analysis of melt rates using satellite remote sensing data, we can only draw conclusions from the basal geometry and its temporal changes using UWB airborne radar."

3. **Surface melting:** You suggest surface melting and the resulting enhanced subglacial discharge could cause enhanced melting. I think this could be improved in two ways. First, I think it would be good to generally discuss how surface hydrology and subglacial hydrology have been found to be linked at several of Greenland's outlet glaciers (e.g., Helheim Glacier), and that a subglacial outflow source for many ice-shelf channels has been hypothesized in Antarctica (e.g., Alley et al., 2016). Second, if there are any indications of surface hydrology in this region in previous studies or satellite imagery you have looked at (e.g., Figure 1b?), that could be useful for further testing this hypothesis.

> We are foremost saying that subglacial discharge has an influence on the melt rates, but it is not as simple as the higher discharge is leading to increased melt. With more subglacial discharge, more freshwater of a so far unclear temperature is brought at an unknown speed into the cavity. There is a clear link between surface water availability and acceleration, with three different patterns of velocity response identified. However, there are no direct measurements of the subglacial discharge. We also do not think that the situation at Helheim (or other tidewater glaciers) is comparable to the situation on a floating tongue glacier. We have channels at the lower side of the floating ice, in which the discharged freshwater may reside and separate the warm ocean masses from the ice base. This would lead to a reduction in melt rates. We can definitely elaborate more on studies of supraglacial lakes in this area, like the studies of Schröder et al., 2020, Neckel et al. 2020 and Hochreuther et al., 2021. It is also worth noting, that Schröder and co-workers found supraglacial lake drainage in winter - this alters the seasonality of availability of subglacial discharge further.

> We added the following sentences to the discussion (Line 381 – 385):
> *"Subglacial discharge has a seasonal component, including supraglacial lake drainage. The drainage of supraglacial lakes is taking place on a short time scale, even within only one day (Neckel et al., 2020). The lag between lake drainage and discharge across the grounding line is not well known, but it is reasonable to assume that the subglacial hydrological system is buffering water. Drainage of supraglacial lakes is not restricted to the summer period, as the study of Schröder et al. (2020) also detected events in winter, which affects the timing of subglacial discharge."*

4. **Appendix D:** This Appendix is really only mentioned in passing in the discussion section, but describes some numerical calculations of ocean currents that are able to support the high melt rates. Consider including this material directly in a new results section (and/or the discussion) along with an explanatory/results figure if you are going to include it in the paper, which you absolutely should in my opinion if it helps explain the ApRES melt rates.

> Thanks for this feedback. We think including the main part of Appendix D in the Discussion section (where this Appendix was referenced before) is a great idea. We will keep the method part (the equation and the description of the three-equation system) in the Appendix D section and reference this in the discussion.

> We moved the following sentences from Appendix D to the discussion (Line 326 – 336):
> *"To melt ice at a rate of 140 m $a^{-1}$ requires a heat flux between 1360 and 1600 W $m^{-2}$ (see Appendix D) depending on the range of the glaciers temperatures which we assume to be between ~ 0 K (temperate ice) and 30 K below the pressure melting point. This heat flux must be provided by the water in the cavity below 79NG. We assume a salinity of 34.5 psu and an ice draft of 320 m, estimated for the location of ApRES2a, where the highest melt rates of 140 m $a^{-1}$ were determined during winter. Measurements of the inflow temperatures exceeded 1.2 °C at the calving front (Schaffer et al., 2020), corresponding to 2.9 K above the pressure melting point at the position of the observation. In order to produce a sufficiently high turbulent heat flux into the boundary layer for this given temperature, an ambient velocity of 0.22 m $s^{-1}$ is required for temperate ice and 0.26 m $s^{-1}$ for ice of 30 K below the pressure melting point (see Appendix D). Previously simulated velocities of a buoyant plume rising along the ice base of 79NG indicate velocities of up to 0.22 m $s^{-1}$ (Reinert et al., 2023). From these numbers, we conclude that the ocean currents underneath 79NG are able to supply a heat flux that is high enough to explain even the highest observed annual mean melt rates if they get in contact with the ice base."*

*Specific comments (minor):*

1. Line 5: I think you should include something about how the highest melt rates are spatially localized (i.e., later you say 95% quantile) and short duration here.

> We agree that it is important to mention the short duration of 17 days and we will adjust the sentence accordingly. However, we think that without mentioning the 95% quantil, the sentence is easier to understand and correctly reflects the measurement result.
> The new sentence may read as (Line 5):
> "Our results show extreme basal melt rates exceeding 150 m $a^{-1}$ over a period of 17 d within a distance of 5 km from the grounding line, where the ice has thinned by 42% since 1998."

> Done.

2. Line 30: "Bentley et al. (2023) gives evidence that the AIW...": suggest saying that this evidence comes from an epishelf lake.

   Done.

3. Line 35: describe how meltwater alters fjord circulation (Straneo et al., 2016 ref)?

   We will adjust the sentence as follows (Line 39):
   "However, the supply of fresh water from glacial surface melting has been found to alter circulation in fjords and basal melting of glaciers by increasing buoyancy-driven circulation and decreasing shelf-forced circulation (Straneo et al., 2016)."

   Done.

4. Line 105: Please clarify what "ice base – ice surface – ice base multiple" means

   We will rewrite this sentence and the two following to make this clearer. The new text will be (Line 134):
   "In order to identify nadir and off-nadir returns, we used the first multiple reflections from the ice base, which were characterized by twice the two-way travel time since they originated from the reflections at the ice base, the ice surface, and again at the ice base. Here we assume that the multiple is strongest for the nadir reflection since, in the case of a flat ice surface, most of the reflected energy from a far-off-nadir reflection is reflected in the opposite direction."

   Done.

5. Equation 4: Define the vertical coordinate system somewhere, i.e., z is in (0,R), but what exactly do 0 and R mean?

   Thanks! 0 m is the depth of the surface and R is the range of the basal return. We will make this clearer.

   We added both to the description of the equation.

6. Figure 1: For a while, I thought that there was a red star near ApRES2, but I see now that it is a black star with a red dot in it. I think labelling the 2a and 2b endpoints on the map would help alleviate any confusion.

   Thanks, we will do so!

   Done.

7.  Line 185: "This can differ from the melt rate in the normal or vertical direction at the basal reflector." I got caught up on this statement, can you explain this in a little more detail?

> There are different possibilities to define the melt rate. In the case of a flat ice base, the measured nadir melt rate is equal to the melt rate in the normal and vertical directions. For an inclined ice base, the measured nadir melt rate is equal to the melt rate in the vertical direction, which is different from that in the normal direction. A measured off-nadir melt rate can differ from both the melt rate in the normal and in the vertical direction. We will add this to the manuscript.
>
> We added this in line 218 – 221.

Related, in Appendix A you say "the resulting basal melting in the vicinity of the measurement is always underestimated, although the nadir melt rate might be lower", and I didn't completely understand that either.

> The first off-nadir basal reflection in the first and the repeated measurement can have occurred at two different locations (locations "A" and "B"). If this is the case, we can conclude that the melt rate has been higher at location B than at location A as otherwise, the first basal reflection would have occurred at location A in both measurements. However, when we compare the range to A and to B, we know the true change in ice thickness at B has been higher. Thus, we underestimate the thinning and the melt rate. This is shown in Fig. A1. If the second basal return occurred at an off-nadir angle, the estimated melt rate is below the vertical melt rate at that location. Still, the nadir melt rate can be even lower, but we cannot determine this melt rate.
>
> We will add this to the revised version of the manuscript.
>
> Done.

8.  Figure B1-B3: I think Including one of these in the main text would be good for understanding the ApRES data/method. I think plotting all of the components you use to calculate the melt rate ($\Delta R$, $\Delta Rs$, and $\Delta Re$) in panel c would be good, along with the melt rate you already have in panel d.

> We have created a new figure to include it in the method section of the main part. This figure includes the components used for melt rate calculation.

[Figure]

Figure caption: Analysis of ApRES1 time series. (a) Time-echogram of a Lagrangian measurement at ApRES1 recorded between August 2016 and June 2022. In 2016 and 2017, several ApRES malfunctions caused data gaps. The black outline marks the first 50 m below the basal return. (b) Mean vertical displacement of englacial segments (dots). The gray shaded area marks the range between the 25% and 75% quantile. Segments between 20 m and 20 m above the first basal return at the end of the measurement period (red dots) were used to calculate the change in ice thickness due to vertical strain by fitting a linear function (black line). (c) Time series of ablation rate (negative for ablation). The grey shaded area marks the uncertainty due to the off-nadir correction. (d) Time series of the determined melt rate (color) within the first 50 m below the basal return, corresponding to the area marked by black lines in (a). (e) Time series of basal melt rate. The dashed line shows the 95% quantile, the solid line the median, and the shaded area marks the range between the 25% and 75% quantile.

Done.

9. Equation (7): I don't entirely understand how you are calculating this in practice but I think the previous comment would help clarify.

Equation 7 deals with the quantification of the ablation. The vertical displacement of all segments from the surface to above the ice base are affected by ablation and strain deformation. Thus, we use the vertical displacement time series of a segment at 50 m depth and correct for the strain in the upper 50 m. The result of Equation 7 is the ablation, $\Delta R_s^n$.

10. Figure 4: I would remove the word "sketch" from the caption as it makes it sound like you are drawing something rather than plotting data

> Yes, you are right. We remove the word sketch here.

> Done.

11. Figure 5: It is hard to see the BedMachine profile in this panel b (is it absent?). Also should probably include BedMachine citation in the caption

> Thanks for noticing that BedMachine was not cited in the caption. The BedMachine profile was absent in (a) and (b). We will add the reference and show the geometry from BedMachine in (a) and (b).

> We added the BedMachine geometry to (a) and (b).

12. Line 225: Which figure are you referring to in Appendix B2 regarding small strains?

> We are referring to all four figures, since at all ApRES sites, the ice thickness change due to strain is small compared to the high melt rates we found near the grounding line. We will make clear, that we refer to the ApRES measurements.

> Done.

13. Line 230: "marker shape of the off-nadir thinning rates" add "in Figure 1" here to clarify

> Thanks, we will add "in Figure 1c".

> Done.

14. Fig 6a: Is there a negative melt rate/freezing towards the right or just zero?

> The calculated median melt rate is about -1 m/a. However, the ApRES data give no indication of basal freezing. The indication would be a sudden decrease in basal amplitude by a few dB. Since the time series at ApRES1 shows no such decline from January 2021 on, we expect the melt rate to be near zero. One reason for the higher uncertainty is the time-mean vertical displacement since the strain at the end of the time series might differ from the time-mean value.

15. In the discussion, I think some of the results concerning basal ice slopes could potentially be connected to some recent studies on the relation between basal ice slope (e.g., "terracing"; Dutrieux et al., 2014) and melt rates (Schmidt et al., 2023; Watkins et al., 2021). For example, on Line 205 you say "With decreasing basal slopes inside the channel, the melt rate also decreases", which is related to these ideas.

> Thanks for raising this point! Yes, indeed, we can broaden that point and we will connect the results to further studies in the revised version. This is a very good idea!

> We added the following sentences (Line 361 – 364):
> *"High-resolution measurements of the basal topography at Petermann and Thwaites glaciers using underwater vehicles have revealed steep-sided terraces and heterogeneous melting (Dutrieux et al., 2014; Schmidt et al., 2023). Since the UWB airborne radar does not allow us to resolve the base in a similar resolution, we consider only the average basal slope and thus interpret the average melt pattern."*

16. Line 337: I wasn't sure what you meant by "because they exceed such melt rates, which are necessary for a steady-state ice thickness"—I found this sentence confusing.

> A Lagrangian basal melt rate changes the ice thickness at a spatial location and thus the basal slope when the basal melt rates are above (or below) the equilibrium value necessary to maintain the ice thickness (and slope). If the melt rate increases, the basal slope gets steeper and thus the ice thickness is not in a steady-state.

> We will split this sentence into two:
> *"These high melt rates of >100 m a$^{-1}$ are caused by thick ice that is in contact with the warm water masses at the bottom of the cavity. Since these melt rates are above those required for a steady-state ice thickness, this leads to ice thinning in Eulerian perspective and thus a steeper base slope."*

> Done.

17. Line 338: "off the center"... center of the glacier? Suggest rewording

> Thanks! We will follow your suggestion.

> Done.

18. Appendix B1: On Line 370, what is β?

> Thanks for spotting this! The last part of the sentence should have been removed as the equation has been changed before submission.
>
> The new sentence will be:
>
> "The estimation of $\Delta R_\varepsilon$ in the case of an off-nadir reflection requires the quantification of the normal and shear components."
>
> Done.

19. Equation B2: Are the shear terms neglected in the z integral in equation B1 to derive equation B2?

> Yes, indeed. B2 is an approximation of the vertical term in B1. We have changed the text to:
>
> "For a nadir reflection ($\alpha$ = 0) where x0 = 0 and y0 = 0, we assume that shear terms are negligible."
>
> Done.

20. Appendix E: If you need to shorten the paper, I did not think this was strictly necessary.

> We agree that the Figure in Appendix E is not essential for the manuscript and would leave the decision to the Editor.

21. Figure 7/Discussion: The surface temperature seems to drop slightly between 2005-2009 period and later years. Could this somehow be related to the decrease in melt rates? In general, more discussion of why the melt rates might be decreasing would be good. I know you say something about the "inflow of colder water", but could a diminishing subglacial outflow due to less surface melt also contribute?

> It is true that there is a slight drop by ~0.3 K in skin temperature, especially between 2010–2014 and 2015–2019. However, as we do not have any data, neither thinning nor surface lowering, before 2010, we cannot discuss this at all. We also agree that a change in subglacial outflow, is likely to change basal melt rates within the channels, but we lack observational data over a sufficiently long time period.

22. Related to previous, you suggest a "recent inflow of colder water", just wondering if there are there any other observations available that might support this idea?

> An ocean-temperature time series exists in front of 79NG from September 2016 to September 2017 that has been used in a publication by Schaffer et al., 2020 (https://doi.org/10.1038/s41561-019-0529-x). This time series showed an increase in temperature until September 2017. There is an extension of this time series until April 2021 that shows a decrease in temperature since January 2018. This data set has not yet been published. However, it confirms our conclusion of the inflow of colder water into the cavity below 79NG.

23. Table A1: In Case D, I was not sure what "simple measurements" meant

> "Simple measurements" means here a point measurement without an antenna array, that does not allow a spatial analysis like the pRES measurement with one receiving antenna. To distinguish between Case B and D, the geometry of the glacier or the location of the reflection needs to be known (e.g., from airborne or swath radar measurements).
> We understand that this is a not well-formulated sentence. We will reformulate this as follows:
> "This type can not be distinguished from Case B without known geometry."
>
> Done.

24. In the introduction, you talk about how basal melting may be related to ice shelf stability or disintegration. I think you should at least mention something about the stability of this system, and the uncertainties in that in the discussion. For example, do you think the channel is going to eventually break through the ice shelf thickness or otherwise destabilize the system somehow? Or, is it all very uncertain given the temporal dynamics of the melt-rate decreasing and possibly complex interactions with ice flow, ocean currents, and atmospheric changes?

> We fully understand that the reviewer wants us to discuss this - it is actually the point we are most interested in, too. We are giving here some of our thoughts on this, but as we do not have robust means to assess this, it would remain in the field of speculation.
>
> Do we think if the channel will break through the surface? It is very difficult to imagine that it will break through by fracture. We would imagine viscoelastic response to take place and eventually also new cracks forming at the surface parallel to the channel. We could not find any evidence for this at the moment. A basal crevasse forming may depend a lot on an initial crack existing there and as the channel has been there now since already a while (month-years), it is unlikely that the current changes in load situation will create a basal crack. If there are any initial, more tiny, basal cracks existing, the high melt was smearing out or melting out a sharp 'notch', making a crack propagation less

likely. If there would be an intersection with the ice surface, it is most likely happening by melting from our perspective.

Based on the data we present in this study, we are expecting that the channel will grow upstream and it may alter the grounding line location. With that it would follow the trend we measured over the past years. Over the next years, this could then also be investigated with more interferometric data. Unfortunately, the area is very challenging for deploying instruments. Otherwise, it would be great to have GPS and ApRES installed upstream of the current channel location to monitor changes with these methods, too.

Would we think that this can lead to a disintegration of the floating tongue? Given that the floating tongue is confined from the sides, even when the channel may break-through (or melt through), it is hard to imagine that this will lead to a disintegration of the tongue. The local stress situation will change, it would basically be a calving front stress condition then, which may be creep-shut over time again. Comparing this to the Brunt Ice Shelf, which is very heterogeneous, but still stable, it may be a melange that would form 'inside' the floating tongue.

Our plan is to survey the profiles used in this study in an upcoming airborne campaign with the same sensors in 2024. This way, we will achieve the right dataset to assess the situation better. In these flights, the aircraft will also carry a high-resolution optical sensor, which would enable us to find newly formed surface cracks, too.

Another approach we are considering is to survey the channel geometry more densely than in the previous airborne campaigns in 2024 and then to conduct a viscoelastic modelling study, too. This may allow us to figure out, which drivers would be needed to achieve a break-through.

As we will remain in the area of speculation, we think it is useful to address this with a sentence in the manuscript, as the reader may also just ask him/herself, what will this mean for the stability (Line 409):
"However, based on our findings of thinning and upstream progression of subglacial channels, we cannot assess their impact on future stability.
It would require numerical models, as well as longer observational time series to evaluate the stability of 79NG and the Northeast Greenland Ice Stream which should be addressed in further studies."

Done.

*Technical corrections:*
1. Line 40: In the last sentence of the paragraph, I suggest reversing the order of clauses (i.e., "Other methods must be used to monitor...")

Done.

2. Line 165: Suggest changing "which results in an underestimated melt rate" to "underestimates the melt rate by X m/yr..." or similar. As written, I thought you meant that 2.7 m/yr was the absolute melt rate, not the underestimation amount.

Thanks. We will rewrite this sentence also due to the comment from Reviewer 2. The new sentences will be (Line: 196):

"The largest $\Delta R_\varepsilon^n$ was found to be 2.7 m for $\Delta t = 1$ a at ApRES2b. In case the change in ice thickness is based on an off-nadir basal reflection, the correction with the nadir range shift due to ice deformation underestimates the melt rate by $\leq 2.7$ m a$^{-1}$."

Done.

3. Line 180: Change (Vaňková et al., 2021) to Vaňková et al. (2021)

Done.

References:
· Alley, K. E., Scambos, T. A., Siegfried, M. R., & Fricker, H. A. (2016). Impacts of warm water on Antarctic ice shelf stability through basal channel formation. *Nature Geoscience*, 9(4), 290-293.
· Chartrand, A. M., & Howat, I. M. (2020). Basal channel evolution on the Getz Ice Shelf, West Antarctica. Journal of Geophysical Research: Earth Surface, 125(9), e2019JF005293.
· Chartrand, A. M., & Howat, I. M. (2023). A comparison of contemporaneous airborne altimetry and ice-thickness measurements of Antarctic ice shelves. Journal of Glaciology, 1-14.
· Dutrieux, P., C. Stewart, A. Jenkins, K. W. Nicholls, H. F. J. Corr, E. Rignot, and K. Steffen (2014), Basal terraces on melting ice shelves, Geophys. Res. Le]., 41, 5506–5513, doi:10.1002/ 2014GL060618.
· Schmidt, B. E., Washam, P., Davis, P. E., Nicholls, K. W., Holland, D. M., Lawrence, J. D., ... & Makinson, K. (2023). Heterogeneous melting near the Thwaites Glacier grounding line. Nature, 614(7948), 471-478.
· Watkins, R. H., Bassis, J. N., & Thouless, M. D. (2021). Roughness of ice shelves is correlated with basal melt rates. Geophysical Research Letters, 48(21), e2021GL094743.
· Wearing, M. G., Stevens, L. A., Dutrieux, P., & Kingslake, J. (2021). Ice-shelf basal melt channels stabilized by secondary flow. Geophysical Research Letters, 48(21), e2021GL094872.

**Authors point-to-point responds Referee Comment #2 to egusphere-2023-1320**

Please find the author's responses in blue below the reviewer's comments.
Please find the implemented changes in red below the responses.

Review of
**Extreme melting at Greenland's largest floating ice tongue**
by Zeising et al., submitted to *The Cryosphere*
The manuscript presents an analysis of a combination of measurements for detecting the local balance conditions in the grounding line region of Nioghalvfjerdsfjorden Glacier (79NG) in NE- Greenland. For this purpose the authors use pRES and ApRES and ultra-wide-band airborne radar measurements, complemented by surface elevation models, derived from TanDEM-X imagery. The geometry data are also compared to earlier seismic investigations form 1998, in order to analyse the long-term changes of the grounding zone of 79NG. They find extremely high melt rates on a local scale, but still considerable strong subglacial melting across the entire grounding zone.

The manuscript is clear and well written and presents a detailed analysis of data quality and comparison of data from different sources. Data and results are very well presented. In general, the results are based on a rigorous processing and analysis approach and provide new insight into the recent and medium term evolution of the grounding zone of 79NG. The manuscript will add important new knowledge to the scientific efforts of understanding the complex interaction of ice, ocean and climate in NE Greenland.

Apart from some minor issues, which I list further down, there is only one major question concerning the localised detection of incised channels into the underside of the glacier. The strong increase of the channel height is documented by UWB radar data between 2018 and 2021 and on a longer time scale by pronounced and locally concentrated surface lowering from SAR imagery. It was stated the measurements of Mayer et al. (2000) show no indication of subglacial channels close to the grounding line in 1998. However, the seismic measurements were performed with a 24 channel instrument, covering horizontal distances of 240 m. The single measurements were up to 2 kilometres apart and the final figure in Mayer et al. (2000) only shows an interpolated cross profile of the single shots. Therefore, it cannot be concluded that there was an absence of subglacial channels in 1998. There exists an unpublished data set of airborne radar data also from 1998 (named the "Niels Reeh data set" in e.g. Seroussi et al., 2011) which shows a much more detailed ice bottom topography in the grounding line region of 79NG. The figure shows a cross profile in the vicinity of the BB' profile, with large subglacial undulations across the entire glacier, where the deepest reaches more than 200 m.

[Figure]

Fig.: RES cross profile in the grounding line region of 79NG from 1998. The dark blue line shows the profile location in Lat/Lon. The orange and bright blue lines show the ice geometry, while the yellow dot indicates the location of the subglacial channel identified in the recent manuscript.

Therefore, I highly recommend to consult to airborne RES data from 1998, in order to reach to sensible conclusions with regard to the temporal evolution of localised melt features.

We fully understand the concern that the seismic data do not reflect the exact geometry due to the measurement setup and we are aware that the measurement locations were separated by up to 2 km. The statement that the seismic data from Mayer et al. (2000) show no evidence of an existing channel relies primarily on one measurement located 350 m from the center of the channel, which had a width of 1 km in 2021. At this measurement location, the largest ice thickness of the across ice-flow line was found.

We agree that the ice thickness distribution based on the "Niels Reeh data set" shows a more detailed ice bottom topography. To our knowledge, this data set is an ice-thickness data set. Therefore can you please advise what the accuracy of the surface elevation data set is? Many thanks in advance.

We really appreciate that the reviewer provided the figure showing the ice geometry of this unpublished data set. Indeed, the figure shows a channel-like feature at the base, which is shifted by a few kilometers from the location of the channel in our study. Similar to the seismic data from Mayer et al (2000), the RES data show almost the largest ice thickness and the deepest draft at the location of today's channel.

To address the concerns of the Reviewer, we will rewrite the sentence as follows: "Geophysical observations provide no evidence that this channel existed in 1998 (Mayer et al., 2000; Seroussi et al., 2011)."

We got access to Reeh's airborne data set and analyzed the ice geometry in and across the flow line. In contrast to the seismic data, the airborne radar data show a channelized ice base at the D-D' profile (5 km downstream from the grounding line) with a channel of 120 m in height where the large channel was found in 2018 and 2021. The profile at the grounding line shows no channel, and the profile at 2.5 km downstream from the grounding line shows only one small channel.

[Figure]

Figure: Across ice flow geometry from Reeh's airborne radar data at the grounding line (GL), 2.5 km downstream and 5 km downstream.

We have changed the according sentence again to (Line 249):
*"Airborne radar data from 1998 conducted by Niels Reeh and Erik Lintz Christensen with the DTU (Technical University of Denmark) Space 60 MHz ice sounder show no channel near the grounding line and a small 120 m high channel, located 5 km downstream the grounding line (Fig. 5d)."*

Additionally, we now use Reeh's airborne radar data to calculate Eulerian thinning rates at D-D' instead of the seismic data and updated the values in the following sentence (Line: 252):
"Compared to 1998, the ice thickness 5 km downstream from the grounding line has decreased by more than 162 m or 32%. Above the large subglacial channel, the glacier thinned by 67%."

*Minor comments:*
L. 51: I might be useful to already mention the data source of the DEMs here (e.g. from InSAR processing)

Thanks, we will add "[…] from bistatic TanDEM-X SAR interferometry […] ".

Done.

L 52: GROCE needs a reference.

We added the link to the webpage of the project.

Done.

L. 53: either "an UWB radar", or the "UWB radar" and adding some information.

We agree and will write "AWI's ultra-wideband (UWB) radar".

We wrote (Line: 73):
"[...] AWI's ultra-wideband (UWB; Alfred Wegener Institute Helmholtz Centre for Polar and Marine Research, AWI) radar [...]"

L. 68: the spatial adjustment requires some error estimate.

We will add a sentence about the error estimate of the spatial adjustment. This is based on the standard deviation of the difference to the laser scanner data over bedrock from 2018.

We added (Line 88):
"The final DEMs were geocoded and spatially adjusted to the global TanDEM-X DEM by calculating the standard deviation and the normalized median absolute deviation (NMAD) for all DEMs over stable ground (Nuth and Kääb, 2011; Wessel et al., 2016). The NMAD is considered to be more robust to outliers than the standard deviation (Höhle and Höhle, 2009). For the entire stack of DEMs we obtain a standard deviation of 0.96 m and an NMAD of 0.55 m which is in agreement with the TanDEM-X mission requirements (Wessel et al., 2018)."

L. 80: I do not understand this sentence. What do you presume in the hardware?

Thanks for finding this. I think this was a typo and should be "pre-summed". We will additionally improve the wording and will use "pre-stacked" instead.

Done.

L. 85: The processing steps change the ground resolution, which should be discussed here.

We will add a sentence about the resolution of the ground resolution after SAR focusing.

We added the following sentence (Line 108):
"The SAR focus is set to achieve a ground resolution of 10 m in the along-track direction."

L. 90: Accuracy of the laser measurements?

The vertical accuracy of the laser scanner DEM is 0.1 m. We will add this to the sentence.

Done.

L. 105: The off-nadir reflections depend on the location of the instrument. This could be demonstrated in more detail here.

We will add the following sentences to demonstrate this:
"The steeper the basal gradient between the nadir ice base and the location of the off-nadir reflection increases on average, the earlier the off-nadir reflection occurs. Thus, if an off-nadir reflection appears before the nadir basal return depends on the location of the measurement relative to the surrounding basal slopes and their gradients."

Done.

L. 135: This is unclear to me. Does that apply generally to single-repeat pRES measurements, or is this a special case?

In general, single-repeated measurements do allow melt rate measurements. However, the water-saturated surface influences the signal for deeper layers in a way that only low correlation values can be found, which prevents the strain-rate analysis and consequently the melt rate analysis.

L. 167: You state that you are able to estimate the $\Delta Rn$, why are you underestimating the melt rate then?

To make this clearer, we will rewrite this sentence as follows:
"The largest $\Delta R_\varepsilon^n$ was found to be 2.7 m for $\Delta t = 1$ a at ApRES2b. In case the change in ice thickness is based on an off-nadir basal reflection, the correction with the nadir range shift due to ice deformation underestimates the melt rate by $\leq 2.7$ m a$^{-1}$."

Done.

L. 180: citation format needs change

Done.

L. 188: The inaccuracy of signal propagation speed does not depend on the melt rate. If you would like to state that the uncertainties in the propagation speed result in similar inaccuracies as about 1% of the melt rate do, this should be reformulated.

Thanks! We will reformulate this sentence as follows:
"A further uncertainty arises from the inaccuracy of the signal propagation speed in the ice resulting in an inaccuracy of the melt rate of ~1% (Fujita et al., 2000)."

Done.

L. 195/196: To which width does the region of surface lowering reduce in which distance?

The width decreases to 500 m within 5 km distance. We will add this to the sentence. Thanks!

We modified the sentences as follows (Line 230):
"Its width decreases in flow direction from a maximum of 1 km roughly 2 km upstream from the grounding line to 500 m within 5 km in ice flow direction (Fig. 1c)."

L. 201: In which distance free floating occurs and what are the criteria for "free flotation"?

The ice is freely floating between 4 and 5 km from the grounding line (depending on the location). This is shown by the lower flexure limit in Fig. 1. The lower flexure limit is based on interferometry which showed that the ice is freely moving up and down with the tides downstream of this limit.
We will add this sentence as follows:
"Five kilometers downstream of the grounding line behind the lower flexure limit where the ice is freely floating (Fig. 1c), […]."
Additionally, we will explain the origin of the upper and lower flexure limit in the caption of Fig. 1.

Done.

L. 222: The Lagrangian perspective also tells only one side of the story. Only the combination completes the information.

Yes, we agree. We will add this to the sentence:
"However, because the ice is flowing, considering the Lagrangian perspective in addition to the Eulerian is necessary for a full understanding of the process that causes these changes."

Done.

L. 222: I would be good state again that the profiles are taken from Fig. 1

Done.

L. 260: The 42% thinning are restricted to a narrow region, compared to the 79NG total extent. This should be mentioned here.

We agree that this sentence was not well expressed. We will rewrite this sentence following your suggestion:
"[...] ."

We have updated this sentence based on the comparison with Reeh's airborne data (Line 298):
"An analysis of the change in 79NG's geometry between 1998 (Reeh's airborne radar) and 2021 (this study) reveals a thinning by 32% in a narrow region 5 km from the grounding line and an ice base that became channelized, especially in the vicinity of the grounding line (Fig. 5)."

L. 270/271: This is true for significant changes in general, but applies also for warmer temperatures and therefore enhanced melt rates.

Indeed! This is why we wrote 'due to the warming of the ocean and atmosphere.'

L. 281: As long as the pinning points exist at the front, seasonal changes cannot be expected.

Yes, that is correct. We decided that it would make sense to delete this sentence.

Done.

L. 307/308: The low melt rates upstream of the grounding line and outside the large channels depend on what? Is there a patchy grounding line, or do you expect a distributed drainage system, if you refer to low water columns?

We think the wording of the sentence was misleading. We expect low melt rates outside of channels to occur (1) upstream of the grounding line and (2) downstream where a low water column exists.
We expect that the subglacial water is drained via the channels and therefore don't expect water outside the channels upstream of the grounding line. Downstream the grounding line, a low water column thickness might exist at some areas where we observed low melt rates.
We will rewrite these sentences as follows:
"This results in an upstream shift in the melt pattern compared to the outside of the channel:
(i) Upstream the grounding line and downstream where a low water column exists, higher melt rates occur inside the channel than outside.
(ii) In the vicinity of the grounding line, where the ice is in contact with warm ocean currents, lower melt rates occur in the channel than outside."

Done.

Seroussi, H., Morlighem, M., Rignot, E., Larour, E., Aubry, D., Ben Dhia, H., & Kristensen, S. S. (2011). Ice flux divergence anomalies on 79north Glacier, Greenland. Geophysical Research Letters, 38(9).

---

## Author Response (AR2)

**Author's response to egusphere-2023-1320**

Dear Editor, Dear Joe MacGregor,

We are pleased that the manuscript has been accepted. We agree that the manuscript would benefit from a chapter on hydrostatic imbalance. Therefore, we have replaced Appendix E with a section about the hydrostatic imbalance near the grounding line of 79° North Glacier, as suggested by the reviewer. We referenced Appendix E in the discussion but made no further changes here.

Many thanks for your efforts to improve our manuscript!

Best wishes
Ole and co-authors